# Learning to Reward: A Contextual Bandit Framework for Distributional Reward Policy Optimization

## Abstract

Reward models (RMs) are a cornerstone of aligning large language models (LLMs) with human preferences, yet their ability to faithfully represent the nuance and uncertainty of these preferences remains a critical challenge. Existing RMs can be classified by their output (point-estimate vs. distributional) and training paradigm (supervised vs. reinforcement learning). While prior work has addressed three of these four quadrants, no existing framework combines the dynamic optimization of reinforcement learning with the rich, uncertainty-aware representation of a distributional model. To fill this gap, we propose **D**istributional **R**eward **P**olicy **O**ptimization (**DRPO**). DRPO formulates the learning of a distributional reward model as a contextual bandit problem, where the RM itself acts as a stochastic policy. This policy is trained using an uncertainty-aware meta-reward signal derived directly from the statistics of the reward distributions. We provide an algorithm analysis of DRPO's gradient dynamics and conduct extensive experiments to demonstrate its effectiveness. Our code and data are available at https://anonymous.4open.science/r/DRPO/.

## 1 Introduction

The alignment of Large Language Models (LLMs) with human values is critically dependent on the quality of reward models (RMs), which serve as proxies for human preferences during Reinforcement Learning from Human Feedback (RLHF) (Christiano et al., 2017; Bai et al., 2022b; Ouyang et al., 2022; Lambert, 2025). An RM is tasked with scoring model-generated responses, providing the essential learning signal that guides the LLM towards helpful, harmless, and honest behavior (Bai et al., 2022a; Dong et al., 2024). The efficacy of the entire alignment process, therefore, hinges on the fidelity and robustness of the underlying reward model (Liu et al., 2025a; Zhong et al., 2025).

To systematically analyze the landscape of existing reward modeling techniques, we categorize them along two orthogonal axes: (1) their output representation, which can be a single or multi-dimensional scalar (*point-estimate*) (Dong et al., 2023; Yuan et al., 2024; Wang et al., 2024a;b; Zhong et al., 2024; Gu et al., 2024) or a full probability distribution (*distributional*) (Wang et al., 2025c; Sun et al., 2025b; Lou et al., 2024); and (2) their optimization paradigm, based on either *Supervised Learning (SL)* (Ye et al., 2025; Ankner et al., 2024; Yu et al., 2025a) or *Reinforcement Learning (RL)* (Liu et al., 2025b; Guo et al., 2025; Chen et al., 2025). This taxonomy, illustrated in Table 1, reveals a critical and underexplored quadrant in the design space of reward models.

The most prevalent approaches reside in the supervised, point-estimate quadrant, where a model is trained to predict scalar or vector-valued rewards (Luo et al., 2025; Zhang et al., 2025b). A common approach is to append a value head to the LLM's final-layer representations to output a reward, optimized either through a ranking objective based on the Bradley-Terry model (Yuan et al., 2024; Yang et al., 2024; Pitis et al., 2024; Sun et al., 2025a) or a direct regression objective (Wang et al., 2024a;b; Zhong et al., 2024). Recently, generative reward models (Mahan et al., 2024; Gu et al., 2024; Saha et al., 2025) have emerged, which leverage the LLM's own generative capabilities to produce a sentence containing the reward score (Ankner et al., 2024; Zhang et al., 2025a; Zhao et al., 2025; Liang et al., 2025). A notable subset of these are Reasoning RMs (Lu et al., 2025; Liu et al., 2025b), which reframe reward prediction as a reasoning task, generating a chain-of-thought

Table 1: A Taxonomy of Reward Modeling Paradigms. Our work, DRPO, is the first to explore the promising but previously unoccupied quadrant of distributional reward modeling via reinforcement learning.

|  | Point-Estimate | Distributional |
|---|---|---|
| **SL** | Bradley-Terry (Pitis et al., 2024; Sun et al., 2025a) Regression-based (Wang et al., 2024a;b) Generative RM (Mahan et al., 2024; Zhao et al., 2025) | Gaussian (Sun et al., 2025b) Categorical (Wang et al., 2025c) No Prior (Dorka, 2024) |
| **RL** | Reasoning RM (Liu et al., 2025b; Guo et al., 2025) | **DRPO (This Work)** |

explanation before outputting the final score to enhance generalization and interpretability (Guo et al., 2025; Chen et al., 2025). Despite these architectural variations, all point-estimate models share fundamental limitations. First, a single vector, $r \in \mathbb{R}^d$, struggles to capture the intricate and diverse nature of human preferences Poddar et al. (2024); Lanchantin et al. (2025). Second, these models cannot express predictive uncertainty, making them susceptible to the noisy and conflicting preference data (Padmakumar et al., 2024; Lou et al., 2024).

To overcome these issues, another line of work has focused on distributional models (Siththaranjan et al., 2024; Wang et al., 2025c; Sun et al., 2025b). These methods explicitly model human preferences as a probability distribution, allowing them to capture uncertainty and better handle data ambiguity. Existing approaches can be sub-categorized by their prior assumptions about the preference distribution, such as assuming a Gaussian (Lou et al., 2024; Yan et al., 2024; Sun et al., 2025b), a Categorical (Wang et al., 2025c), or an implicit distribution (Furuta et al., 2024; Dorka, 2024). While providing a richer representational capacity, these methods have so far been confined to the supervised learning paradigm.

This leaves a crucial gap in the literature: a framework that combines the dynamic, exploratory benefits of reinforcement learning with the rich, uncertainty-aware representation of distributional models. To fill this gap, we introduce **D**istributional **R**eward **P**olicy **O**ptimization (**DRPO**), the first framework, to our knowledge, to address this challenge. We formulate the training of a distributional reward model as a meta-level contextual bandit problem, where the RM itself is a stochastic policy that acts by assigning a reward distribution to a given response. This policy is optimized using a novel, uncertainty-aware meta-reward signal designed to distinguish true preference signals from statistical noise. The entire process is structured as a two-stage curriculum that ensures stable and effective learning, progressing from a general prior to a refined, expert model. Our algorithm analysis shows that DRPO's policy gradient is adaptively scaled by the model's predictive uncertainty, providing a more robust and better-calibrated learning signal than conventional point-estimate approaches. In summary, our contributions are:

- We propose DRPO, a two-stage RL framework that formulates distributional reward modeling as a contextual bandit problem, filling a critical gap in the existing literature.

- We design an uncertainty-aware meta-reward function that provides a statistically principled and robust gradient signal, accounting for the inherent stochasticity of the reward distributions.

- We provide an algorithm analysis of DRPO's gradient dynamics and conduct extensive experiments to demonstrate its effectiveness.

## 2 RELATED WORK

### 2.1 POINT-ESTIMATE REWARD MODELS

Point-estimate reward models represent the predominant approach to learning from human preferences. These models are trained to output a single or multi-dimensional scalar score (Luo et al., 2025; Zhang et al., 2025b) for a given response, i.e., $r_\phi : (x, y) \to \mathbb{R}^d$. Classic methods append a value head to a language model and optimize it using objectives derived from the Bradley-Terry model (Dykstra, 1960; Yuan et al., 2024; Yang et al., 2024; Pitis et al., 2024; Sun et al., 2025a), learning directly from pairwise preference data (Dykstra, 1960; Yuan et al., 2024). Another com-

mon technique is to frame the task as a regression problem, training the model to predict explicit scores provided by human raters (Dong et al., 2023; Wang et al., 2024a;b; Zhong et al., 2024). More recently, generative RMs (Mahan et al., 2024; Ankner et al., 2024; Gu et al., 2024) have been proposed, which leverage the innate capabilities of LLMs to produce the reward score textually (Zhang et al., 2025a; Zhao et al., 2025; Liang et al., 2025; Wang et al., 2025b;a; Yu et al., 2025a; Whitehouse et al., 2025). This has been extended to Reasoning RMs (Liu et al., 2025b; Hong et al., 2025), which treat reward prediction as a reasoning task. These models first generate a chain-of-thought explanation for a preference before outputting the final score, which has been shown to improve generalization and interpretability (Guo et al., 2025; Chen et al., 2025; Lu et al., 2025; Whitehouse et al., 2025; Ning et al., 2025; Yu et al., 2025b). Despite their success and diversity, all point-estimate RMs share a fundamental limitation: their low-dimensional scalar output is an impoverished representation of complex, and often ambiguous, human preferences. They lack an inherent mechanism to represent the model's predictive uncertainty, making them susceptible to noise in the preference data and leading to poorly calibrated rewards.

## 2.2 Distributional Reward Models

To address the shortcomings of point-estimate RMs, a second line of work has focused on distributional reward modeling (Siththaranjan et al., 2024). The core idea is to model the reward for a given response not as a single number, but as a full probability distribution, $r_\phi : (x, y) \rightarrow P(\mathbb{R})$, thereby capturing uncertainty and a richer representation of preference (Wang et al., 2025c; Yang et al., 2025b; Yao et al., 2025). Existing works can be categorized by their distributional assumptions. Some approaches model the reward distribution with a parametric form, such as a Gaussian distribution to capture mean and variance (Lou et al., 2024; Yan et al., 2024; Sun et al., 2025b), or a Categorical distribution over a discrete set of rating buckets (Wang et al., 2025c). Others employ non-parametric methods (Furuta et al., 2024), such as quantile regression (Dorka, 2024), to learn the distribution with fewer prior assumptions. While these approaches successfully integrate the concept of uncertainty into reward modeling, they have so far been confined to the SL paradigm. In this setting, the model learns a static mapping from inputs to reward distributions based on a fixed dataset. This leaves a critical gap in the literature: the development of a distributional reward model within an RL framework. DRPO is the first work to address this gap.

## 3 Preliminaries

**Point-Estimate Reward Models.** A conventional point-estimate reward model, $r_\phi(x, y)$, is trained on a preference dataset $\mathcal{D} = \{(x_i, y_i^w, y_i^l)\}_{i=1}^N$. The parameters $\phi$ are typically optimized by minimizing the Bradley-Terry (BT) negative log-likelihood objective (Dykstra, 1960):

$$\mathcal{L}_{\text{BT}}(\phi) = -\mathbb{E}_{(x, y^w, y^l) \sim \mathcal{D}} \Big[ \log \sigma \big( r_\phi(x, y^w) - r_\phi(x, y^l) \big) \Big]$$

**Group Relative Policy Optimization.** Our framework employs Group Relative Policy Optimization (GRPO) (Shao et al., 2024) for training. GRPO is a variant of PPO that updates a policy $\pi_\theta$ by maximizing a clipped surrogate objective, regularized by a KL-divergence penalty against a reference policy $\pi_{ref}$. Its learning signal is a group-relative advantage, $\hat{A}_k$, calculated by standardizing rewards within a sampled batch of size $G$:

$$\hat{A}_k = \frac{R_k - \mu_R}{\sigma_R}, \quad \text{where} \quad \mu_R = \frac{1}{G} \sum_{j=1}^G R_j, \text{ and } \sigma_R = \sqrt{\frac{1}{G} \sum_{j=1}^G (R_j - \mu_R)^2}.$$

The full objective function is:

$$\mathcal{J}_{\text{GRPO}}(\theta) = \mathbb{E}\left[ \frac{1}{G} \sum_{k=1}^G \min\Big( \rho_k(\theta)\hat{A}_k, \, \text{clip}_{1\pm\gamma}(\rho_k(\theta))\hat{A}_k \Big) - \beta \mathcal{D}_{\text{KL}}(\pi_\theta \| \pi_{ref}) \right],$$

where $\rho_k(\theta)$ is the probability ratio $\pi_\theta(a_k|s)/\pi_{\theta_{\text{old}}}(a_k|s)$.

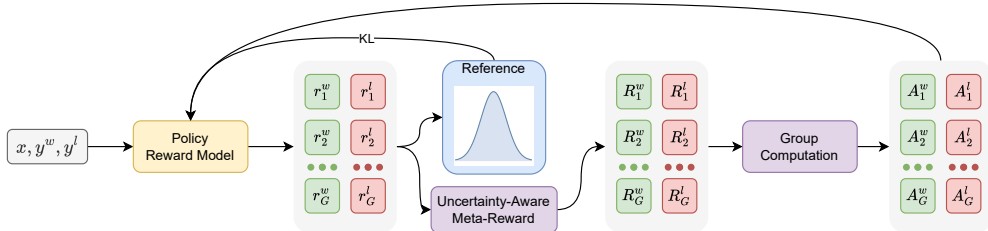

Figure 1: Overview of the DRPO framework. The policy reward model, $r_\phi$, is treated as a stochastic policy that generates reward samples for a preference pair $(y^w, y^l)$. These samples are evaluated by uncertainty-aware meta-reward function to produce robust signals, $R_k$. The meta-rewards are then standardized into group-relative advantages, $\hat{A}_k$, which update the policy via a KL-regularized objective against a reference policy, $\pi_{ref}$.

## 4 DISTRIBUTIONAL REWARD POLICY OPTIMIZATION

In this section, we introduce distributional reward policy optimization (DRPO), a two-stage framework that trains reward models by extending the reinforcement learning with verifiable reward (RLVR) (Su et al., 2025) paradigm to the preference learning. The core of our approach is to **reframe reward model learning as a policy optimization task**. This reframing hinges on addressing two fundamental questions: 1) If the reward model functions as a policy model, what is the nature of its output? 2) What signal serves as the "reward of reward" (or meta-reward) to guide the optimization of this policy?

As illustrated in Figure 1, we formulate the training of the reward model as a contextual bandit problem. This is a stateless reinforcement learning paradigm defined by the following components:

- Context Space $\mathcal{C}$. A context $c \in \mathcal{C}$ is a prompt-response pair $(x, y)$ that requires evaluation.
- Action Space $\mathcal{A}$. An action $r \in \mathcal{A}$ is the reward value assigned to the context, where the action space is the set of real numbers, $\mathcal{A} = \mathbb{R}$.
- Policy $\pi_\phi$. The parameterized RM, $r_\phi$, serves as the agent's stochastic policy $\pi_\phi(r|c)$. It maps a given context $(x, y)$ to a probability distribution over actions (reward values).
- Reward Function $\mathcal{R}$. The environment, encapsulating the preference dataset $\mathcal{D}$, provides a meta-reward. For actions $a^w$ and $a^l$ taken in contexts corresponding to a preference pair $(y^w \succ y^l)$, the environment returns a positive reward if the actions satisfy the preference, i.e., $r^w > r^l$.

### 4.1 DISTRIBUTIONAL REWARD POLICY MODEL

As framed in Section 2, our approach conceptualizes the reward model as a policy that can be either deterministic, yielding a scalar reward, or stochastic, yielding a reward distribution. We argue that the stochastic formulation aligns more naturally with the principles of policy optimization. In this view, the action space corresponds to the continuous domain of possible reward values, and the RM-as-policy, denoted $\pi(r|x, y)$, represents a probabilistic belief over these values for a given input. Consequently, we adopt a distributional reward model as our policy. Specifically, we employ an RM with a probabilistic value head that models the human preference for a response $(x, y)$ as a gaussian distribution, parameterized by its mean and variance: $r_\phi(x, y) \sim \mathcal{N}(\mu, \sigma^2)$. This distributional choice enables explicit uncertainty quantification. While a scalar output provides a single, absolute judgment, a distribution captures both the expected reward ($\mu$) and the model's confidence in that assessment ($\sigma^2$). This allows the model to express ambiguity for controversial inputs rather than collapsing to a single, potentially misleading score.

### 4.2 TWO-STAGE REINFORCEMENT LEARNING

Our proposed framework, DRPO, utilizes GRPO as the policy optimization algorithm within a two-stage training process that embodies a curriculum learning strategy. The first stage, which we term

**DRPO-Zero**, initiates training from a base model. The primary objective of this stage is to learn a general-purpose and well-behaved foundational reward metric. Subsequently, the second stage, **DRPO-Refine**, initializes its model using the parameters learned in DRPO-Zero. We then increase the difficulty of the learning task in this stage, compelling the model to acquire a more fine-grained and discerning reward function. Algorithm 1 is provided in Appendix B.

**DRPO-Zero** The objective of this initial stage is to train a distributional reward model, $r_\phi$, to learn a foundational reward distribution, starting from a base model. The policy is parameterized to output a Gaussian distribution, $r_\phi(x, y) \sim \mathcal{N}(\mu, \sigma^2)$. We initialize the policy, $r_{\phi_0}$, such that its output approximates a standard normal distribution for any given input. This choice establishes an unbiased prior, reflecting the assumption that the initial model has no inherent preference before training. This distribution is also used as the fixed reference policy for this stage, $\pi_{ref} \sim \mathcal{N}(0, 1)$. To generate a learning signal, we sample $G$ rewards for both the preferred and dispreferred responses and evaluate the quality of each sample using the Uncertainty-Aware Meta-Reward, as formally described in Definition 1.

**Definition 1 (Uncertainty-Aware Meta-Reward)** *Given two sets of $G$ reward samples, $\mathbf{r}^w = \{r_i^w\}_{i=1}^G$ from the distribution $r_\phi(x, y^w) \sim \mathcal{N}(\mu^w, (\sigma^w)^2)$ and $\mathbf{r}^l = \{r_j^l\}_{j=1}^G$ from $r_\phi(x, y^l) \sim \mathcal{N}(\mu^l, (\sigma^l)^2)$, the meta-reward $R_k$ for a sample $r_k \in \mathbf{r}^w \cup \mathbf{r}^l$ is defined as:*

$$R_k = \begin{cases} \frac{1}{G} \sum_{j=1}^G \mathbb{I}(r_k^w > r_j^l + m_{kj}) & \text{if } r_k \in \mathbf{r}^w \\ \frac{1}{G} \sum_{j=1}^G \mathbb{I}(r_j^w > r_k^l + m_{jk}) & \text{if } r_k \in \mathbf{r}^l \end{cases},$$

*where $\mathbb{I}(\cdot)$ is the indicator function. The dynamic margin $m_{ij} = K \cdot \sqrt{(\sigma_i^w)^2 + (\sigma_j^l)^2}$ is proportional to the standard deviation of the difference between the two corresponding reward distributions, with $K$ serving as a confidence-controlling hyperparameter.*

The inclusion of the dynamic margin $m_{ij}$ is critical for creating a robust learning signal. A naive reward based on simple numerical comparison, $\mathbb{I}(r^w > r^l)$, is highly susceptible to statistical noise. The key challenge is to determine whether an observed difference $r^w > r^l$ represents a true preference signal or is merely a product of random fluctuation from the sampling process. Our intuition is that a genuine signal is present only when this difference significantly exceeds the inherent

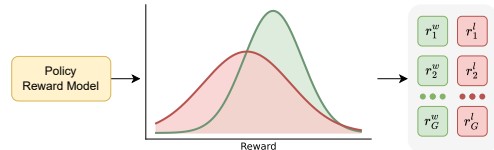

Figure 2: Sample from estimated distribution.

stochasticity of the comparison. We formalize this by considering the distribution of the difference $d = r^w - r^l$. Since the rewards are sampled from independent Gaussians, their difference also follows a Gaussian distribution: $d \sim \mathcal{N}(\mu^w - \mu^l, (\sigma^w)^2 + (\sigma^l)^2)$. The standard deviation of this difference, $\sigma_{diff} = \sqrt{(\sigma^w)^2 + (\sigma^l)^2}$, perfectly quantifies the magnitude of the expected random fluctuation, or "noise". Therefore, by setting the margin proportional to $\sigma_{diff}$, our meta-reward only credits comparisons that are statistically significant, yielding a more stable and reliable training signal.

With the vector of meta-rewards $\mathbf{R}$ computed for all $2G$ samples, we calculate the group-relative advantages by standardizing these values: $\hat{A}_i = (R_i - \mu_\mathbf{R})/\sigma_\mathbf{R}$. The policy $r_\phi$ is then updated by maximizing the GRPO objective. This is a on-policy policy gradient objective and includes a KL penalty against the fixed prior:

$$\mathcal{J}(\phi) = \mathbb{E}_{\substack{(x,y^w,y^l)\sim\mathcal{D} \\ r\sim r_\phi}} \left[ \frac{1}{2G} \sum_{i=1}^{2G} \left( \log \pi_\phi(r_i|x, y_i) \cdot \hat{A}_i - \beta \mathcal{D}_{\text{KL}}[\pi_\phi(\cdot|x, y_i)\|\mathcal{N}(0, 1)] \right) \right], \quad (1)$$

where the policy probability $\pi_\phi(r_i|x, y_i)$ is the value of the probability density function (PDF) of the learned Gaussian $\mathcal{N}(\mu_i, \sigma_i^2)$ evaluated at the sample $r_i$.

**On-Policy Sampling and Objective Simplification.** We note that while standard GRPO and PPO implementations for language models rely on importance sampling and clipping to maintain stability, these components are not strictly necessary in our DRPO framework. In standard RLHF,

autoregressive token generation is computationally expensive, necessitating the reuse of generated trajectories for multiple gradient updates. This introduces a discrepancy between the behavior policy ($\pi_{old}$) and the current policy ($\pi_\theta$), requiring the importance ratio $\rho = \pi_\theta/\pi_{old}$ and clipping to correct for off-policy drift. In contrast, DRPO formulates reward modeling as a stateless bandit problem where the "action" is a single scalar sampled from a Gaussian. Since sampling a scalar value is computationally negligible, we sample fresh reward actions strictly *on-policy* at every optimization step. Consequently, the importance ratio remains $\rho = 1$, rendering the clipping mechanism redundant. This allows us to simplify the optimization to a direct policy gradient update without the need for complex off-policy corrections.

**DRPO-Refine**  In this second stage, we aim to refine the foundational model obtained from DRPO-Zero to yield a more precise and accurate reward distribution. The training procedure largely mirrors that of DRPO-Zero, with two critical modifications that constitute a curriculum learning approach.

First, the initialization and the reference policy are updated. While DRPO-Zero required a simple prior ($\mathcal{N}(0,1)$) due to the absence of a well-formed starting point, the model resulting from the first stage, which we denote $r_{\phi_{zero}}$, provides a strong foundation. Consequently, we initialize both the policy model $\pi_\phi$ and the reference model $\pi_{ref}$ with the parameters of $r_{\phi_{zero}}$. The policy model's parameters are subsequently updated during training, whereas the reference model's parameters remain frozen, serving as a stable anchor for fine-tuning.

Second, we increase the training difficulty to challenge the already capable model. This is achieved by adjusting two key hyperparameters. We increase the confidence-controlling hyperparameter in the meta-reward to $K' > K$, making it more difficult for the policy to earn a positive meta-reward. Concurrently, we use a larger KL divergence coefficient, $\beta' > \beta$, to enforce a stronger regularization towards the now-competent reference policy. The optimization objective remains the same in form:

$$\mathcal{J}(\phi) = \mathbb{E}_{\substack{(x,y^w,y^l)\sim\mathcal{D} \\ r\sim r_\phi}} \left[ \frac{1}{2G} \sum_{i=1}^{2G} \left( \log \pi_\phi(r_i|x,y_i) \cdot \hat{A}_i - \beta' \mathcal{D}_{KL}[\pi_\phi(\cdot|x,y_i)\|\pi_{ref}(\cdot|x,y_i)] \right) \right], \quad (2)$$

where $\pi_{ref}$ is now the frozen model $r_{\phi_{zero}}$.

## 4.3 Algorithm Analysis

We provide a analysis of DRPO by dissecting the gradients that drive parameter updates and comparing them to those of the conventional Bradley-Terry (BT) model. This reveals how DRPO's formulation inherently addresses the limitations of uncertainty-agnostic, point-estimate reward modeling.

**The Bradley-Terry Gradient**  The BT model seeks to find parameters $\phi$ by minimizing the negative log-likelihood of the observed preferences:

$$\mathcal{L}_{BT}(\phi) = -\log \sigma(r_\phi(y^w) - r_\phi(y^l))$$

The gradient with respect to $\phi$ is:

$$\nabla_\phi \mathcal{L}_{BT} = -\left(1 - \sigma(r_\phi^w - r_\phi^l)\right) \cdot (\nabla_\phi r_\phi^w - \nabla_\phi r_\phi^l) = -\sigma(r_\phi^l - r_\phi^w) \cdot (\nabla_\phi r_\phi^w - \nabla_\phi r_\phi^l)$$

The gradient pushes $r_\phi^w$ higher and $r_\phi^l$ lower, as expected. However, the update's magnitude is scaled by $\sigma(r_\phi^l - r_\phi^w)$, a term entirely dependent on the scalar difference between the rewards. This has two key limitations: 1) it is agnostic to the model's confidence in its own predictions, treating a high-confidence prediction and a lucky guess identically if their reward difference is the same; and 2) it is less tolerant of noisy data. Suppose there is a preference for flipping labels, i.e., flipping the labels of $y^w$ and $y^l$. It will still optimize in the wrong direction.

**The DRPO Policy Gradient**  DRPO maximizes the objective $\mathcal{J}(\phi)$ using a policy gradient method. The policy $\pi_\phi$ is the distributional RM, which we parameterize as a Gaussian, $r_\phi(x,y) \sim \mathcal{N}(\mu_\phi(x,y), \sigma_\phi(x,y)^2)$. The gradient of the objective relies on the score function, $\nabla_\phi \log \pi_\phi(r|s)$,

which can be decomposed with respect to the distribution's parameters:

$$\nabla_{\mu_\phi} \log \pi_\phi(r|s) = \frac{r - \mu_\phi(s)}{\sigma_\phi(s)^2} \tag{3}$$

$$\nabla_{\sigma_\phi} \log \pi_\phi(r|s) = \frac{(r - \mu_\phi(s))^2 - \sigma_\phi(s)^2}{\sigma_\phi(s)^3} \tag{4}$$

The full policy gradient is thus $\nabla_\phi \mathcal{J}(\phi) \propto \mathbb{E}_{r \sim \pi_\phi}[\hat{A} \cdot \nabla_\phi \log \pi_\phi(r|s)]$, where $\hat{A}$ is the advantage derived from uncertainty-aware meta-reward.

**Comparative Analysis.** This formulation provides two fundamental advantages over the BT model. First, it enables uncertainty-aware gradient scaling for the mean. The gradient for the expected reward, $\mu_\phi$, is scaled by the inverse variance, $1/\sigma_\phi^2$. This acts as an adaptive learning rate intrinsic to the model. When the model is uncertain about a reward (high $\sigma_\phi$), the gradient's magnitude is suppressed, preventing large, potentially unstable updates. Conversely, when the model is confident (low $\sigma_\phi$), the gradient is amplified, promoting more decisive learning. Second, it allows for principled uncertainty learning. DRPO provides a direct learning signal for the uncertainty parameter, $\sigma_\phi$. When rewarded samples (i.e., those with high $\hat{A}$) are close to the mean, the term $((r - \mu_\phi)^2 - \sigma_\phi^2)$ becomes negative, pushing $\sigma_\phi$ to decrease and thus increasing the model's confidence. When rewarded samples are far from the mean, this term becomes positive, pushing $\sigma_\phi$ to increase, thereby reflecting learned uncertainty for ambiguous or noisy preferences. This mechanism for explicitly learning confidence is entirely absent in point-estimate frameworks.

## 5 EXPERIMENT

### 5.1 EXPERIMENTAL SETUP

All DRPO models are trained on the Skywork-Reward-Preference-80K-v0.2 dataset (Liu et al., 2024a). To ensure our findings are not specific to a single architecture, we use three distinct backbone models: Llama-3.1-8B (Grattafiori et al., 2024), Qwen3-4B, and Qwen3-8B (Yang et al., 2025a). Following the taxonomy, we compare DRPO against baselines from the three other established quadrants: (1) Supervised Learning (SL) point-estimate methods, including Offset-Bias (Park et al., 2024), ArmoRM (Wang et al., 2024b), Skywork-RM (Liu et al., 2024a) and InternLM2 (Cai et al., 2024); (2) SL distributional methods, including URM (Lou et al., 2024) and QRM (Dorka, 2024); and (3) Reinforcement Learning (RL) point-estimate methods, including DeepSeek-GRM (Liu et al., 2025b), RM-R1 (Chen et al., 2025) and J1 (Whitehouse et al., 2025). We evaluate the performance of all trained models on seven established reward model benchmarks: RewardBench (Lambert et al., 2024), RewardBench v2 (Malik et al., 2025), PPE Preference & Correctness (Frick et al., 2024), RMB (Zhou et al., 2024), RM-Bench (Liu et al., 2024b), and JudgeBench (Tan et al., 2024). Further details on all experimental settings, including hyperparameters and implementation specifics, are provided in Appendix C.

### 5.2 OVERALL PERFORMANCE

We present the main evaluation results on seven reward model benchmarks in Table 2. The findings demonstrate the effectiveness of our DRPO framework across different model scales and architectures. Notably, DRPO 4B-scale model achieves a result that is comparable to or exceeds that of several baseline models with significantly larger parameter counts, such as InternLM2-20B and ArmorRM-Llama3-8B , highlighting the parameter efficiency of DRPO. Furthermore, 8B-scale DRPO models deliver highly competitive performance against the strongest baselines from all quadrants. They achieve the best performance on three of the seven benchmarks: PPE Correctness (66.2), RMB (66.9), and JudgeBench (64.9). These results underscore the substantial potential of the reinforcement learning distributional paradigm for training high-fidelity reward models.

### 5.3 ABLATION STUDIES & ANALYSIS

**Meta-Reward Design** The design of the meta-reward signal is critical to the success of DRPO. An intuitive alternative is to directly reward the margin by which preferred reward samples exceed

Table 2: Performance comparison of reward models across seven benchmarks. Bold values indicate the best performance in each benchmark.

| | RewardBench | RewardBench v2 | PPE Pref | PPE Corr | RMB | RM-Bench | JudgeBench | Avg. |
|---|---|---|---|---|---|---|---|---|
| SL point-estimate models | | | | | | | | |
| OffsetBias-Llama3-8B | 89.0 | 64.8 | 59.2 | 64.1 | 57.8 | 71.3 | 63.5 | 67.1 |
| ArmoRM-Llama3-8B | 90.4 | 66.5 | 60.6 | 60.6 | 64.6 | 69.3 | 59.7 | 67.4 |
| Skywork-Llama3.1-8B | **93.1** | 71.8 | **62.2** | 62.5 | 66.6 | 72.1 | 62.3 | **70.1** |
| InternLM2-7B | 87.6 | 53.4 | 62.0 | 60.4 | 67.1 | 67.1 | 56.5 | 64.9 |
| InternLM2-20B | 90.2 | 56.3 | 61.0 | 63.0 | 62.9 | 68.3 | 64.3 | 66.6 |
| RL point-estimate models | | | | | | | | |
| DeepSeek-GRM-27B | 86.0 | - | 64.7 | 59.8 | 69.0 | - | - | - |
| RM-R1-Qwen-7B | 85.2 | - | - | - | 66.4 | 70.2 | - | - |
| J1-Llama-8B | 85.7 | - | 60.3 | 59.2 | - | **73.4** | 42.0 | - |
| SL distributional models | | | | | | | | |
| URM-Fsfairx-8B | 89.9 | 67.2 | 57.7 | 60.9 | 59.6 | 67.1 | 60.3 | 66.1 |
| URM-Skywork-8B | 92.9 | **73.9** | 60.6 | 61.0 | 65.9 | 69.6 | 63.4 | 69.6 |
| QRM-Llama3-8B | 91.2 | - | - | - | 62.4 | - | 61.1 | - |
| RL distributional models | | | | | | | | |
| DRPO-Qwen3-4B | 87.6 | 68.8 | 59.8 | 62.2 | 64.4 | 69.2 | 63.1 | 67.9 |
| DRPO-Qwen3-8B | 89.0 | 70.0 | 61.1 | 64.7 | 63.2 | 70.6 | **66.3** | 69.3 |
| DRPO-Llama3.1-8B | 90.0 | 68.6 | 60.7 | **66.2** | **66.9** | 71.4 | 65.1 | 69.8 |

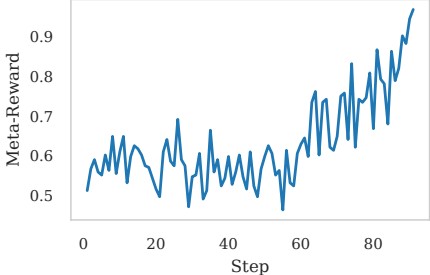

(a) Meta-Reward of hinge-based meta-reward.

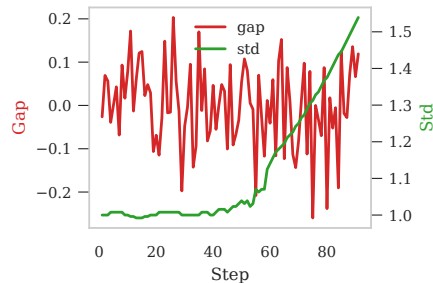

(b) Gap and std of hinge-based meta-reward.

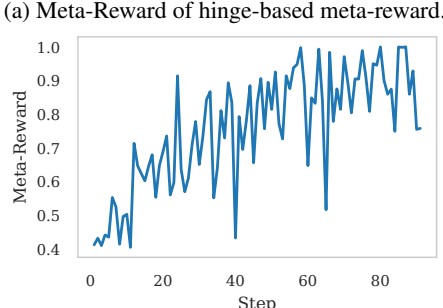

(c) Meta-Reward of uncertainty-aware meta-reward.

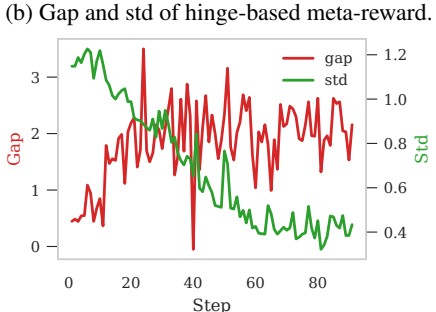

(d) Gap and std of uncertainty-aware meta-reward.

Figure 3: DRPO training dynamics with different meta-reward.

dispreferred ones. We investigate this with a hinge-based meta-reward, defined for a sample $r_k$ as:

$$R_{\text{hinge}}(r_k) = \begin{cases} \frac{1}{G} \sum_{j=1}^{G} \max(0, r_k^w - r_j^l) & \text{if } r_k \in \mathbf{r}^w \\ \frac{1}{G} \sum_{j=1}^{G} \max(0, r_j^w - r_k^l) & \text{if } r_k \in \mathbf{r}^l \end{cases}$$

While this formulation is appealing, we find it is susceptible to reward hacking. As shown in Figure 3, although the expected meta-reward for the hinge-based approach increases monotonically (Figure 3a), this trend is misleading. The mean gap between the preferred and dispreferred reward distributions, $\mu^w - \mu^l$, fails to increase, while the learned standard deviation, $\sigma$, grows pathologically (Figure 3b). This demonstrates a clear failure mode where the policy maximizes its reward by inflating its variance to find favorable outlier samples, rather than learning to improve its core discriminative ability.

In contrast, uncertainty-aware meta-reward exhibits the desired behavior. As shown in Figure 3c and Figure 3d, its meta-reward stably increases while the standard deviation consistently decreases, indicating the model is becoming more confident in its outputs. Concurrently, the mean gap widens, confirming that the model's discriminative power is genuinely improving. This comparison validates that the uncertainty-aware design does not exhibit this vulnerability and leads to stable and meaningful convergence.

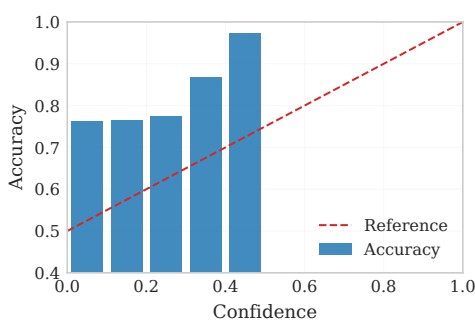
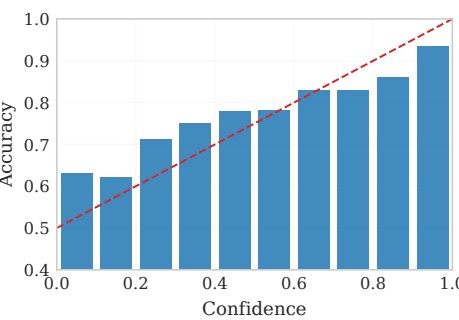

(a) Results from DRPO-Zero stage.

(b) Results from DRPO-Refine stage.

Figure 4: DRPO accuracy against its confidence in RewardBench.

**Reward Calibration and Confidence** A key advantage of distributional reward models is that their output reflects not only preference but also predictive uncertainty. This allows us to analyze the calibration of our models at each stage of the DRPO curriculum. We first formalize the model's confidence for a given preference pair.

**Definition 2 (Sample Confidence)** *For a given preference pair $(y^w, y^l)$, let the output distributions from a reward model be $r_\phi(x, y^w) \sim \mathcal{N}(\mu^w, (\sigma^w)^2)$ and $r_\phi(x, y^l) \sim \mathcal{N}(\mu^l, (\sigma^l)^2)$. The model's confidence in this preference is defined as:*

$$C(x, y^w, y^l) = \left| 1 - 2 \cdot P(r^w > r^l) \right| = \left| 1 - 2 \cdot \Phi\left( \frac{\mu^w - \mu^l}{\sqrt{(\sigma^w)^2 + (\sigma^l)^2}} \right) \right|,$$

*where $P(r^w > r^l)$ is the probability that a sample from the "preferred" distribution is greater than a sample from the "dispreferred" distribution, and $\Phi(\cdot)$ is the cumulative distribution function (CDF) of the standard normal distribution.*

Figures 4a and 4b show the accuracy against its confidence in RewardBench for the models from the DRPO-Zero and DRPO-Refine stages, respectively. The model from the DRPO-Zero stage exhibits a nascent ability to distinguish preferences, but its predictions are poorly calibrated and concentrated in low-confidence regions. This outcome is consistent with the goal of this initial stage: to learn a well-regularized, foundational reward distribution that serves as a robust starting point, rather than to produce a perfectly calibrated final model. In contrast, the DRPO-Refine model demonstrates strong calibration. Its predictive confidence (Definition 2) closely tracks its empirical accuracy, as shown by its alignment with the ideal calibration line in Figure 4b. This indicates the model learns to reliably estimate its own uncertainty. A low confidence score corresponds to a low probability of being correct, providing a valuable signal for identifying challenging or out-of-distribution inputs.

Additionally, the confidence score allows us to categorize the training data into easy (high-confidence, correct), boundary (low-confidence), and hard (high-confidence, incorrect) samples. By using rejection sampling to increase the proportion of boundary and hard examples, we can focus the fine-tuning process on the most informative data. As shown in Figure 5, the distribution of preference probabilities, $P(r^w > r^l)$, scored by the DRPO-Zero model is bimodal. One peak near 0.7 represents data

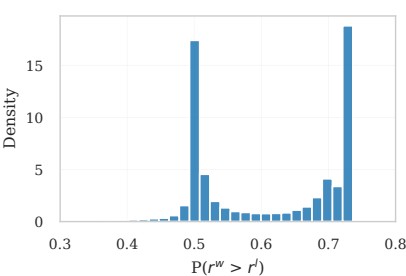

Figure 5: Training data scored by DRPO-Zero.

the model can distinguish with some confidence, while the other peak near 0.5 (random chance) signifies data where the model is highly uncertain. This aligns with the observations from Figure 4a.

## 6 CONCLUSION

In this work, we addressed the limitations of conventional reward models by introducing DRPO, a framework that learns a distributional reward model via reinforcement learning. By formulating the task as a contextual bandit problem, our method treats the reward model as a stochastic policy and optimizes it with a principled, uncertainty-aware meta-reward signal. Our algorithm analysis demonstrates that DRPO's gradients are adaptively scaled by the model's predictive uncertainty, while our empirical results show that it produces well-calibrated reward models that outperform baselines from across the reward modeling landscape.

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

## A  USE OF LARGE LANGUAGE MODELS

Throughout the preparation of this manuscript, we utilized LLMs to assist in several aspects of the research process. Specifically, LLMs were employed to improve the clarity and readability of the text by correcting grammatical errors and refining sentence structures. Additionally, we used LLMs as a tool to aid in the generation of boilerplate code for parts of our experimental implementation. All final code, experimental designs, and written content were reviewed, validated, and finalized by the authors.

## B  METHOD

The full training process is shown in Algorithm 1.

---

**Algorithm 1** Distributional Reward Policy Optimization (DRPO)

---

**Require:** Preference dataset $\mathcal{D}$; base model for initialization; hyperparameters $K, \beta$ for Stage 1; $K', \beta'$ for Stage 2; number of samples per response $G$.

    **Stage 1: DRPO-Zero (Prior-Guided Pre-training)**
1: Initialize policy $r_\phi$ from the base model.
2: Set the fixed reference policy $\pi_{ref}(\cdot) \leftarrow \mathcal{N}(0, 1)$.
3: **for** each training iteration **do**
4:     Sample a batch of preferences $\{(x, y^w, y^l)\} \sim \mathcal{D}$.
5:     **for** each preference pair in the batch **do**
6:         Get reward distributions: $r_\phi(x, y^w)$ and $r_\phi(x, y^l)$.
7:         Sample reward sets $\mathbf{r}^w = \{r_i^w\}_{i=1}^G$ and $\mathbf{r}^l = \{r_j^l\}_{j=1}^G$ via reparameterization.
8:         Compute meta-rewards $\mathbf{R} = \{R_k\}_{k=1}^{2G}$ for all samples using Def. 1 with hyperparameter $K$.
9:     **end for**
10:    Compute group-relative advantages $\hat{\mathbf{A}}$ by standardizing the collected meta-rewards $\mathbf{R}$.
11:    Update parameters $\phi$ by ascending the gradient $\nabla_\phi \mathcal{J}(\phi)$ (as defined in Eq. 1).
12: **end for**
13: Store the resulting parameters $\phi_{zero} \leftarrow \phi$.

    **Stage 2: DRPO-Refine (Self-Refinement)**
14: Initialize policy $r_\phi \leftarrow r_{\phi_{zero}}$.
15: Initialize and freeze the reference policy $\pi_{ref} \leftarrow r_{\phi_{zero}}$.
16: (Optional) Curate a refined dataset $\mathcal{D}' \leftarrow \text{ActiveLearning}(\mathcal{D}, r_{\phi_{zero}})$ using Def. 2.
17: **for** each training iteration **do**
18:    Sample a batch of preferences from $\mathcal{D}'$ (or $\mathcal{D}$).
19:    ... (Repeat lines 6-9, using the increased hyperparameter $K'$ for the meta-reward).
20:    Compute group-relative advantages $\hat{\mathbf{A}}$.
21:    Update parameters $\phi$ by ascending the gradient $\nabla_\phi \mathcal{J}(\phi)$ (as defined in Eq. 2, using $\beta'$ and $\pi_{ref}$).
22: **end for**
**Ensure:** The refined distributional reward model $r_\phi$.

---

### B.1  DISCUSSION ON RL FORMULATION: CONTEXTUAL BANDITS VS. MDPS

In this section, we provide a clarification regarding the classification of DRPO within the Reinforcement Learning landscape, specifically distinguishing our Contextual Bandit formulation from the

standard Markov Decision Process (MDP) often utilized in language model alignment Ouyang et al. (2022).

**Contextual Bandit vs. MDP** Standard RLHF approaches typically model the *generation* of text as an MDP. in that setting, the state space consists of the current token sequence, the action space is the vocabulary, and the horizon $T$ corresponds to the sequence length. The policy generates a trajectory of tokens, and credit assignment (via value functions) is required to attribute the final reward to intermediate steps.

In contrast, DRPO formulates *reward modeling* as a **Contextual Bandit** problem.

- **Context ($\mathcal{C}$):** The input is the complete prompt-response pair $(x, y)$, which is fully observable and static.
- **Action ($\mathcal{A}$):** The action is the reward score $r \in \mathbb{R}$ assigned to the context.
- **Horizon:** Since the model outputs the reward in a single step, the horizon is effectively $T = 1$.

Consequently, extending DRPO to a full MDP with value function fitting (e.g., $V(s)$) is not applicable, as there are no state transitions or intermediate rewards to model. The "return" is simply the immediate meta-reward received from the environment.

**Exploration and Exploitation in DRPO** Despite the stateless nature of the bandit formulation, the core RL mechanism of the *exploration-exploitation trade-off* remains fundamental to our approach:

- **Exploration:** The reward model, acting as a stochastic policy $\pi_\phi(r|x,y) \sim \mathcal{N}(\mu, \sigma^2)$, explores by sampling diverse reward values (actions) from its distribution. A higher variance $\sigma$ encourages the exploration of a wider range of reward values to discover those that satisfy the meta-reward margin constraints.
- **Exploitation:** The agent exploits by shifting its mean $\mu$ and adjusting its variance $\sigma$ toward regions that yield high meta-rewards (i.e., maximizing the margin between preferred and dispreferred responses) while minimizing the KL-divergence penalty.

**Adaptation of Group Relative Policy Optimization (GRPO)** We further clarify how the GRPO algorithm is adapted from the sequence-level to the sample-level:

- In **Generative GRPO**, the "group" consists of $G$ distinct *token sequences* generated from a single prompt.
- In **DRPO**, the "group" consists of $G$ distinct *reward values* sampled from the policy distribution for a single context $(x, y)$.

Thus, our optimization operates within the continuous action space of rewards. The group-relative advantage is calculated by standardizing the meta-rewards of these samples, providing a robust, low-variance gradient estimate without the need for a learned value function baseline.

## C EXPERIMENT SETTING

### C.1 TRAINING SETTINGS

**Dataset.** We use the Skywork-Reward-Preference-80K-v0.2 (Liu et al., 2024a) dataset for all training runs. It aggregates high-quality open-source data from multiple domains, ensuring diversity across general chat, reasoning (math/code), and safety. The dataset is a composite of four primary sources: HelpSteer2 (Wang et al., 2024c), Magpie Series (Xu et al., 2024), OffsetBias (Park et al., 2024) and WildGuardMix (Han et al., 2024). The diverse mixture ensures that our DRPO framework is evaluated on a distribution that covers both helpfulness (HelpSteer2), robustness (OffsetBias), safety (WildGuardMix), and complex reasoning (Magpie Ultra/Pro). To facilitate our two-stage curriculum, we randomly partition the dataset into a 10% subset for the DRPO-Zero stage and the remaining 90% for the DRPO-Refine stage.

**Training Infrastructure.** All models are trained on a cluster of four nodes, each equipped with eight NVIDIA H-series GPUs. We use a global batch size of 512 and a maximum sequence length of 2048. All models are trained for one epoch with a linear learning rate decay schedule.

**Ablation and Analysis Experiments.** Unless otherwise specified, all ablation studies and analysis experiments presented in the paper are conducted using the Qwen3-4B model as the backbone.

**Hyperparameters.** The key hyperparameters for our DRPO framework are detailed in Table 3. We increase both the KL coefficient and the confidence margin hyperparameter in the second stage to create a more challenging and regularized fine-tuning objective.

Table 3: Key hyperparameters for the DRPO framework.

| Hyperparameter | DRPO-Zero | DRPO-Refine |
|---|---|---|
| Learning Rate | $2 \times 10^{-5}$ | $1 \times 10^{-5}$ |
| KL Coefficient ($\beta$) | 0.05 | 0.1 |
| Confidence Margin ($K$) | 0.05 | 0.1 |
| **Shared Parameters** | | |
| Global Batch Size | 512 | |
| Max Sequence Length | 2048 | |
| Number of Samples ($G$) | 32 | |
| Epochs | 1 | |
| LR Schedule | Linear | |

# D    SUPPLEMENTARY EXPERIMENT

## D.1    EARLY TRIAL: SUPERVISED DISTRIBUTIONAL TRAINING AS STAGE 1

In this section, we present an analysis of an early exploratory experiment that applies supervised distributional training as stage 1. In this experiment, we attempted to train the Gaussian reward head using a Bradley-Terry loss applied directly to the sampled rewards. Specifically, for a given pair $(y_w, y_l)$, we sampled rewards $r_w \sim \mathcal{N}(\mu_w, \sigma_w^2)$ and $r_l \sim \mathcal{N}(\mu_l, \sigma_l^2)$ using the reparameterization trick and minimized the negative log-likelihood of the preference.

**Training Dynamics.** The training curves are visualized in Figure 6.

- **Apparent Success in Ranking:** As shown in Figure 6a and 6b, the model appears to train successfully: the training loss decreases consistently, and the pairwise accuracy approaches 100%. Similarly, Figure 6c shows that the reward gap ($\mu_w - \mu_l$) widens effectively, indicating that the model is learning to distinguish between preferred and dispreferred responses.

- **Failure in Distributional Modeling (Variance Collapse):** However, Figure 6d reveals a critical failure mode inherent to this supervised approach. The standard deviation ($\sigma$) of the predicted distributions rapidly collapses to near-zero values within the first few steps of training.

**Analysis.** This phenomenon, which we term *Variance Collapse*, occurs because the standard BT objective solely incentivizes the correct ranking of samples. To minimize the loss, the optimizer drives the model to become deterministic, eliminating the "noise" (variance) that might lead to a flipped ranking during sampling.

**Conclusion.** The collapse of $\sigma$ effectively degenerates the distributional model into a standard point-estimate model. While it achieves high ranking accuracy, it completely loses the ability to capture predictive uncertainty or perform the calibration shown in our main results (Figure 4). This failure demonstrates that the exploration-exploitation trade-off provided by our contextual bandit formulation (which inherently maintains variance to maximize the meta-reward) is strictly necessary for learning robust and well-calibrated reward distributions.

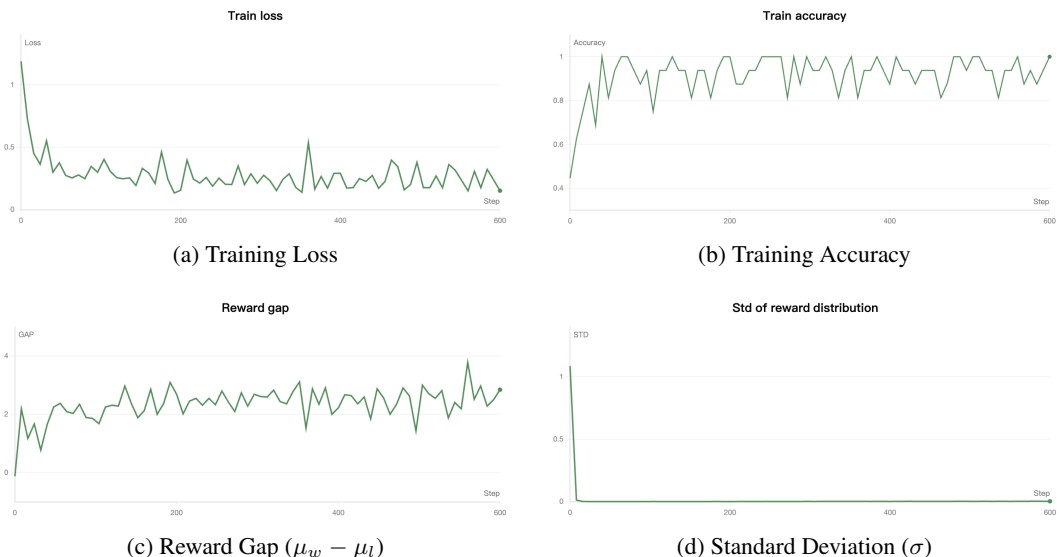

(a) Training Loss

(b) Training Accuracy

(c) Reward Gap ($\mu_w - \mu_l$)

(d) Standard Deviation ($\sigma$)

Figure 6: **Variance Collapse in Supervised Distributional Training.** While the supervised baseline successfully learns to minimize loss (a) and maximize accuracy (b) and reward gap (c), it suffers from catastrophic variance collapse (d). The standard deviation $\sigma$ shrinks to near-zero, stripping the model of its ability to represent uncertainty.

## D.2 SENSITIVITY ANALYSIS OF CONFIDENCE MARGIN $K$

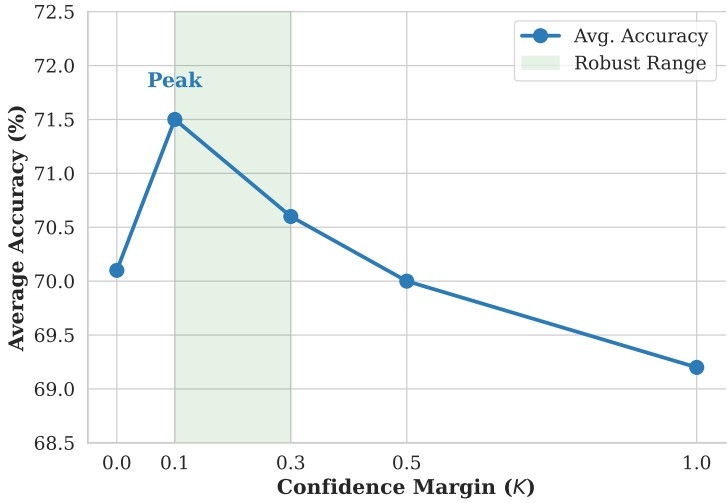

Figure 7: **Ablation study on the confidence margin hyperparameter $K$.**

To validate the design choice of the uncertainty-aware meta-reward, we conducted a sensitivity analysis on the confidence margin hyperparameter $K$. This parameter controls the strictness of the margin $m_{ij} = K \cdot \sqrt{(\sigma^w)^2 + (\sigma^l)^2}$, effectively setting the required signal-to-noise ratio for a preference pair to be considered valid for learning.

**Experimental Setup.** We trained separate DRPO models varying $K \in \{0, 0.1, 0.3, 0.5, 1.0\}$. All other hyperparameters remained fixed as per the main experimental settings. Due to the computational constraints of the rebuttal period, we report the average accuracy across five representative benchmarks (RewardBench, RewardBench v2, RMB, RM-Bench, and JudgeBench), excluding the long-running cost PPE series.

**Results and Analysis.** As illustrated in Figure 7, the performance follows a distinct trajectory that corroborates our theoretical intuition:

- **Baseline Robustness** ($K = 0$)**:** Even with $K = 0$, the model achieves a respectable average accuracy. This confirms that the *Intrinsic Gradient Scaling* derived in Eq. 3 and Eq. 4 provides a baseline level of robustness. Even without an explicit margin, the variance-scaled gradient prevents the model from over-fitting to highly uncertain samples. However, the lack of a margin allows the meta-reward to credit "lucky guesses", preventing the model from reaching peak performance.

- **Optimal Range** ($K \in [0.1, 0.3]$)**:** Performance peaks at $K = 0.1$ and remains robust at $K = 0.3$. By requiring the preference signal to exceed the uncertainty floor ($m_{ij}$), the objective successfully filters out ambiguous comparisons while retaining high-confidence training signals. This confirms that $K$ is not an arbitrary heuristic, but a control variable for the required statistical significance of the reward difference.

- **Over-Regularization** ($K \geq 0.5$)**:** As $K$ increases beyond 0.5, performance degrades. An overly aggressive margin leads to **Signal Sparsity**: the condition $r^w - r^l > m_{ij}$ becomes increasingly difficult for the stochastic policy to satisfy, reducing the frequency of positive meta-rewards and slowing down convergence. Furthermore, forcing an artificially large gap on the training data acts as an excessive prior that may not align with the ground truth preference distribution, harming generalization to unseen test data.

### D.3 GENERALIZATION TO CATEGORICAL DISTRIBUTION

To demonstrate that the effectiveness of the DRPO framework is not reliant on the Gaussian assumption, we conducted an ablation study using a **Categorical distribution**. In this setting, the reward output is modeled as a probability distribution over a discrete set of support values (bins), allowing the model to capture multimodal or asymmetric preference distributions that a Gaussian might miss.

**Implementation.** We replaced the Gaussian head with a classification head that outputs logits over $N$ discrete bins. The policy $\pi_\theta(r|x, y)$ becomes a categorical distribution. The meta-reward calculation and policy gradient updates were adapted to operate on sampled discrete values, maintaining the core logic of the contextual bandit framework.

**Results.** Due to time constraints, we evaluated this variant on five out of the seven benchmarks used in the main paper (excluding PPE Preference and PPE Correctness due to high evaluation latency). We performed minimal hyperparameter tuning for the Categorical variant. The comparison is shown in Table 4.

Table 4: Performance comparison between the standard Gaussian parameterization and a Categorical parameterization of DRPO. Despite minimal tuning, the Categorical variant achieves comparable performance, demonstrating the flexibility of the framework.

| Distribution Type | RewardBench | RewardBench v2 | RMB | RM-Bench | JudgeBench | Avg. |
|---|---|---|---|---|---|---|
| **Gaussian (Ours)** | **87.6** | **68.8** | **64.4** | 69.2 | 63.1 | **70.6** |
| **Categorical** | 82.6 | 65.3 | 62.3 | **69.9** | **65.7** | 69.2 |

**Analysis.** The Categorical model achieves an average score of 69.2, which is highly competitive with the Gaussian baseline (70.6). Notably, the Categorical model outperforms the Gaussian model on *RM-Bench* (+0.7) and *JudgeBench* (+2.6), suggesting that for certain tasks, the increased expressiveness of the categorical distribution may be beneficial. The slightly lower performance on RewardBench may be attributed to the lack of extensive hyperparameter tuning compared to the Gaussian baseline.

Crucially, these results confirm that the **Contextual Bandit formulation** of DRPO is robust and effective regardless of the specific parameterization of the reward distribution. The framework successfully learns to optimize the policy via exploration and exploitation, whether the action space is continuous (Gaussian) or discrete (Categorical).

