# OpenReview forum: "Learning to Reward: A Contextual Bandit Framework for Distributional Reward Policy Optimization"
_ICLR.cc/2026/Conference — Submitted to ICLR 2026_

### Official Review · Reviewer_bz4m · 2025-10-26

**Soundness:** 3
**Presentation:** 3
**Contribution:** 2
**Rating:** 4
**Confidence:** 3

**Summary:**

The paper introduce DRPO (Distributional Reward Policy Optimization). It frames reward-model training as a contextual bandit where the reward model itself is considered as a policy that outputs a distribution over rewards. The reward model is trained by preference-based samples with an uncertainty-aware meta-reward. Experiments evaluate across seven benchmarks, showing competitive results. Certain ablation studies are also included.

**Strengths:**

The proposed method is conceptually new as far as I know. It introduces a principled formulation of reward modeling training within an RL framework. The idea is easy to grasp. The paper also provides comprehensive empirical validation, including comparisons across diverse benchmarks and ablations. Collectively they demonstrate the method’s practicality.

**Weaknesses:**

My main concern is the performance. In Table 2, the proposed RL-based distributional models achieve only comparable results to existing approaches. This raises the question of why one should prefer RL-based distributional methods, given that SL approaches are typically simpler and more computationally efficient to train. (Please correct me if it is wrong)

In addition, the meta-reward formulation appears somewhat heuristic. The motivation behind the specific definition in Definition 1 is heuristic as well as the design choice of $m_{ij}$. While I admit that it is encouraging this design works empircally, a deeper explanation would be helpful.

**Questions:**

The choice of a Gaussian head may be restrictive. It would be desirable to see non-Gaussian heads such as quantiles and categorical distributions.

It would be nice to see some downstream impact directly. People may wonder the performance of the learned policy if it is fine-tuned via the proposed RM.

The framework is bandits. Can we extend it to Markov decision processes by fitting value models? What will be the challenges?

---

> ### Author Response · Authors · 2025-11-21
> **response to Reviewer bz4m (1/3)**
>
> We thank the reviewer for their thoughtful assessment and for recognizing our method as "conceptually new" and "easy to grasp." We appreciate the opportunity to clarify the advantages of our RL-based distributional framework compared to traditional approaches.
>
>
>
> ### **W1: On Performance and the Necessity of the RL Paradigm (vs. Supervised Learning).**
>
> We thank the reviewer for challenging us on this fundamental point. We agree that on standard benchmarks, our method achieves results comparable to, rather than uniformly dominating, existing approaches. However, we respectfully argue that **looking solely at accuracy masks the true advantages of DRPO**.
>
> The choice of an RL-based contextual bandit framework over simpler Supervised Learning (SL) is justified by three critical factors: **Parameter Efficiency**, **Prevention of Variance Collapse**, and **Superior Calibration**.
>
> **1) Parameter Efficiency (Efficiency vs. Scale)**
> While the raw scores are comparable, the *cost* to achieve them differs significantly. DRPO demonstrates exceptional parameter efficiency. As shown in **Table 2**, our **DRPO-Qwen3-4B** model (67.9 avg) competes with and often outperforms significantly larger baselines, such as **InternLM2-20B** (66.6 avg) and **ArmoRM-Llama3-8B** (67.4 avg).
>
> **2) Why RL? The "Variance Collapse" Problem in SL**
> The reviewer asks a vital question: *"Why prefer RL... given that SL approaches are typically simpler?"*
> Our answer lies in the difficulty of training distributional heads via supervision.
>
> *   **The Failure Mode of SL:** In our early experiments attempting to train Gaussian reward models via standard Supervised Learning (e.g., minimizing NLL or BT loss directly on distribution parameters), we frequently observed a phenomenon of **"Variance Collapse."** The model aggressively minimizes the variance ($\sigma \to 0$) to reduce the loss, effectively degenerating into a point-estimate regressor. It fails to capture true epistemic uncertainty. We **have added this early trial and discussion to Appendix D**.
> *   **The RL Solution:** DRPO’s contextual bandit formulation prevents this by treating variance as **exploration**.
>     *   In our framework, the policy $\pi_\theta(r|x,y) \sim \mathcal{N}(\mu, \sigma^2)$ *must* maintain non-zero variance to explore the reward landscape and find values that satisfy the meta-reward margin.
>     *   If $\sigma$ collapses prematurely, exploration stops, and the agent cannot maximize the long-term meta-reward. The RL objective therefore implicitly enforces a healthy level of uncertainty, leading to a model that is not just accurate, but statistically robust.
>
> **3) Superior Calibration**
> This difference in training dynamics results in superior calibration, a critical metric for downstream reliability (e.g., Active Learning, OOD detection). As shown in Figure 4b, **the DRPO model’s predictive confidence closely tracks its empirical accuracy**. In contrast, SL models often suffer from overconfidence due to the variance collapse described above.
>
>
>
> ### **W2: meta-reward formulation.**
>
> The reviewer notes that the meta-reward and margin $m_{ij}$ appear heuristic. We clarify that this formulation is grounded in **Signal-to-Noise Ratio (SNR)** principles rather than arbitrary heuristics.
>
> The core challenge in distributional RL is distinguishing whether a sample $r^w > r^l$ indicates a true preference or random sampling noise.
>
> *   Our dynamic margin is defined as $m_{ij} \propto \sqrt{(\sigma^w)^2 + (\sigma^l)^2}$.
> *   This term effectively represents the standard deviation of the difference distribution $d = r^w - r^l$.
> *   Therefore, requiring $r^w - r^l > m_{ij}$ is statistically equivalent to requiring that the sampled difference exceeds the expected noise floor (similar to a one-sided t-test).

---

> ### Author Response · Authors · 2025-11-21
> **response to Reviewer bz4m (2/3)**
>
> ### **Q1:  Choice of Gaussian Head vs. Quantiles/Categorical**
>
> We chose the Gaussian parameterization for two specific reasons relevant to Policy Optimization:
>
> 1.  **Gradient Dynamics:** The Gaussian distribution provides a closed-form score function $\nabla_\phi \log \pi_\phi$, which allows for the precise analytical decomposition of gradients into mean-shifting and variance-scaling components (Eq. 3 & 4). This makes the optimization dynamics transparent and stable.
> 2.  **Continuous Action Space:** We model the reward as a continuous variable. Categorical distributions require discretizing the reward space (binning), which introduces quantization error and loses ordinal information between bins.
>
> *   We agree that Quantile Regression is a promising direction. However, modeling quantiles as a policy in a bandit setting is non-trivial because sampling from a set of quantiles to compute a policy gradient is less direct than the reparameterization trick available for Gaussians.
>
> **New Experiment with Categorical Distribution:** To address the concern regarding limited expressiveness, we implemented a variant of DRPO using a **Categorical distribution** (modeling the reward as a probability distribution over discrete value bins). Due to the time constraints of the rebuttal, we performed minimal hyperparameter tuning and evaluated on 5 of the 7 benchmarks (excluding PPE tasks due to their high computational cost).
>
> **The results (detailed in Appendix D.3)** are summarized below:
>
> | Model                  | RewardBench |  RB v2   |   RMB    | RM-Bench | JudgeBench | **Avg**  |
> | :--------------------- | :---------: | :------: | :------: | :------: | :--------: | :------: |
> | **DRPO (Gaussian)**    |  **87.6**   | **68.8** | **64.4** |   69.2   |    63.1    | **70.6** |
> | **DRPO (Categorical)** |    82.6     |   65.3   |   62.3   | **69.9** |  **65.7**  |   69.2   |
>
> **Analysis:**
>
> 1.  **Comparability:** Despite limited tuning, the Categorical variant achieves performance comparable to the Gaussian model (Avg 69.2 vs. 70.6), notably outperforming the Gaussian model on RM-Bench and JudgeBench.
> 2.  **Framework Robustness:** This result confirms that t**he effectiveness of DRPO stems from the Contextual Bandit formulation (the exploration-exploitation dynamics), rather than being dependent on a specific distributional assumption.** The framework is flexible enough to accommodate more expressive distributions when needed.
>
> **We have added these results and a detailed analysis to Appendix D.3.**
>
> ### **Q2: Downstream Impact.**
>
> The reviewer asks a valuable question regarding the direct downstream impact of using DRPO for policy fine-tuning. While we did not perform a full PPO training run within the rebuttal window, we respectfully highlight that our evaluation on **RewardBench v2** and **RMB** directly assesses this capability.
>
> Specifically, both RewardBench v2 and RMB explicitly evaluate **Best-of-N (BoN)** performance—the ability of the reward model to correctly identify the highest-quality response among multiple model-generated candidates.
>
> *   **Proxy for Fine-tuning:** This Best-of-N capability is not just a static metric; it is the fundamental operation behind **Rejection Sampling Fine-tuning** and is highly correlated with the signal quality required for **Reinforcement Learning Fine-tuning**. If a reward model can reliably rank a set of $N$ candidates, it provides the necessary gradient signal for policy improvement.
> *   **Empirical Evidence:** As shown in **Table 2**, DRPO demonstrates superior performance on these benchmarks. For instance, on **RewardBench v2**, DRPO-Qwen3-8B achieves a score of **70.0**, significantly outperforming the much larger InternLM2-20B (56.3).

---

> ### Author Response · Authors · 2025-11-21
> **response to Reviewer bz4m (3/3)**
>
> ### **Q3: Extend to Markov decision processe**
>
> The reviewer asks an insightful question about extending the framework to MDPs. **We respectfully clarify that such an extension is not directly applicable, as DRPO is strictly formulated as a stateless Contextual Bandit problem.**
>
> This question implies a comparison to standard RLHF (like PPO/GRPO), where an agent generates tokens sequentially. We wish to explicitly contrast these two paradigms to explain why the MDP formulation does not fit the *reward modeling* task:
>
> *   **Standard RLHF (MDP):** The input is a prompt, and the output is a sequence of tokens. The state changes after every token, creating a trajectory with horizon $T$ (sequence length).
> *   **DRPO (Contextual Bandit):** The input is a complete `(prompt, response)` pair, which serves as the **Context**. The output is a reward score (or distribution), which serves as the **Action**. Since the model outputs the reward in a single step, the horizon is effectively $T=1$.
>
> **Why the Bandit Formulation is Sufficient (Exploration-Exploitation)**
> While we do not extend to an MDP, we emphasize that the core principles of RL, specifically **Exploration** and **Exploitation** are fully active in our Bandit setting:
>
> *   **Exploration:** The agent "explores" by sampling diverse reward values (actions) from its current Gaussian policy $\pi_\phi(r|x,y) \sim \mathcal{N}(\mu, \sigma^2)$. High $\sigma$ encourages exploring the reward space to find values that satisfy the meta-reward margin.
> *   **Exploitation:** The agent "exploits" by shifting its mean $\mu$ and adjusting $\sigma$ toward regions that yield high meta-rewards.
> *   Unlike an MDP which requires credit assignment across time (fitting a Value function $V(s)$ to predict future returns), our task has immediate feedback. The "return" is the immediate meta-reward.
>
> **Clarification on GRPO Implementation**
> To further clarify the distinction, we note how the GRPO logic adapts to this Bandit setting:
>
> *   **In Standard GRPO (MDP):** The "Group" consists of $G$ different **token sequences** generated from one prompt.
> *   **In DRPO (Bandit):** The "Group" consists of $G$ different **reward values** sampled from the policy for the *same* context $(x, y)$.
>     Therefore, our method operates at the **"sample-level"** within the continuous action space of rewards, **rather than the token/sequence-level**. Extending this to an MDP would require defining "state transitions" for a reward score, which is **not semantically meaningful** in the context of preference learning.
>
> We **have revised Appendix B** to explicitly add this discussion.
>
>
>
> We hope that our responses and the corresponding manuscript revisions have addressed your concerns. We welcome any further questions and would be happy to provide any additional clarification needed.

---

> ### Author Response · Authors · 2025-11-27
>
> Dear Reviewer bz4m,
>
> Thank you again for your constructive review and the time you have dedicated to evaluating our paper.
>
> We formally submitted our detailed rebuttal a few days ago. As the discussion period approaches its final week, we wanted to briefly highlight how we have addressed your primary concerns, particularly regarding the necessity of the RL paradigm and the design of the meta-reward. We believe these additions significantly strengthen the paper and address the contribution concern. We would deeply value your feedback on these new results and are happy to answer any remaining questions you might have.
>
> Best regards,
>
> Authors of Paper 9353

---

### Official Review · Reviewer_jXS5 · 2025-10-27

**Soundness:** 2
**Presentation:** 2
**Contribution:** 3
**Rating:** 4
**Confidence:** 3

**Summary:**

This paper introduces Distributional Reward Policy Optimization (DRPO), a novel framework for training reward models (RMs) that combines reinforcement learning (RL) with distributional reward modeling to better capture uncertainty and improve alignment in large language models (LLMs). DRPO treats reward modeling as a contextual bandit problem, where the reward model acts as a stochastic policy. The paper presents both theoretical contributions, including an analysis of DRPO’s gradient dynamics, and practical experiments comparing DRPO to existing baseline models. The results show DRPO's efficacy in producing well-calibrated reward models that outperform traditional approaches.

**Strengths:**

The proposed DRPO framework is a good idea. By treating reward modeling as a contextual bandit problem, the paper provides a perspective on how reward models can be trained dynamically, with the reward model acting as a stochastic policy.

**Weaknesses:**

The paper does not adequately highlight the novelty and distinctions of DRPO in comparison to prior work in reward modeling. While DRPO extends scalar reward learning to a distributional paradigm, it mainly focuses on parameterizing the reward distribution using a Normal distribution [1]. However, the paper could provide a clearer explanation of how this approach improves on existing models.

 **contextual bandit problem :** The paper introduces the contextual bandit formulation but does not clearly explain how it applies to reward modeling. The core idea behind contextual bandits lies in the exploration-exploitation trade-off in a one-step decision environment. However, DRPO does not seem to fully align with this concept. The GRPO objective averages over a sequence, but DRPO does not. Does this mean DRPO's approach can still be classified as a contextual bandit formulation? The paper would benefit from more clarification, especially whether the framework operates at a sequence-level or token-level.

**Limited explanation of Hyperparameters:** While the hyperparameters for DRPO-Zero and DRPO-Refine are mentioned, the paper lacks an in-depth explanation of why these specific hyperparameters were chosen and how they impact model performance. For example, the confidence-controlling hyperparameter $K$ is a critical setting, but its influence on the model's performance is not fully discussed. Is there any sensitivity analysis performed for $K$?

[1]  Probabilistic uncertain reward model.

**Questions:**

1. Could you clarify why the standard normal distribution is chosen as the fixed prior in Equation (1)?

2. I have some confusion regarding the two-stage RL method. Could you explain the difference between Equation (1) and Equation (2)? What motivates and justifies the need for DRPO-Refine? Additionally,  is there a particular reason clipping and importance sampling are not used in the objective function, as it is in GRPO (though I understand it could be beneficial for gradient analysis)?

**Details Of Ethics Concerns:**

No ethical concerns are apparent in the paper.

---

> ### Author Response · Authors · 2025-11-21
> **response to Reviewer jXS5 (1/4)**
>
> We sincerely thank the reviewer for their time and for acknowledging that DRPO provides a "novel framework" and a "good idea" for dynamic reward training. We value the constructive feedback regarding the clarity of our contextual bandit formulation and experimental details. We address the specific concerns below.
>
> ###  **Clarification on the Contextual Bandit Formulation & GRPO**
>
> We thank the reviewer for this insightful comment. We realize that t**he distinction between our reward-generation formulation and standard token-generation RL** was not explicitly contrasted in the paper. We respectfully clarify that the reviewer’s concern stems from viewing the problem through the lens of standard autoregressive generation (Markov Decision Process), whereas our framework is strictly a **stateless Contextual Bandit problem**.
>
> **Difference from Standard RLHF (MDP vs. Bandit):**
>
> *   **Standard RLHF (e.g., PPO for generation):** The input is a prompt, and the output is a sequence of tokens generated autoregressively. This is modeled as a **Markov Decision Process (MDP)** where the horizon $T$ is the sequence length.
> *   **DRPO (Our Framework):** The input is a complete `(prompt, response)` pair, which serves as the **Context**. The output is a reward score (or distribution), which serves as the **Action**. Since the model outputs the reward in a single step, the horizon is $T=1$. This fits the precise mathematical definition of a **Contextual Bandit**.
>
> **the "Exploration-Exploitation" Concern:**
> The reviewer asks if DRPO aligns with the exploration-exploitation concept. **It effectively does.**
> In our framework, the reward model acts as a stochastic policy $\pi_\phi(r|x,y) \sim \mathcal{N}(\mu, \sigma^2)$.
>
> *   **Exploration:** The agent "explores" by sampling diverse reward values (actions) from its current Gaussian distribution. High $\sigma$ encourages exploring a wider range of potential reward values to find those that satisfy the meta-reward condition.
> *   **Exploitation:** The agent "exploits" by shifting its mean $\mu$ and adjusting its variance $\sigma$ toward regions that yield high **Meta-Rewards** (i.e., maximizing the margin between preferred and dispreferred responses).
> *   **The Trade-off:** The model must balance increasing variance (to discover samples that satisfy the margin) against decreasing variance (to minimize the KL penalty and stabilize predictions). Unlike Supervised Learning, which fits a static label, our agent actively searches for a distribution of scores that maximizes the return.
>
> **the "Sequence-Level vs. Token-Level" (GRPO) Concern:**
> The reviewer correctly notes that GRPO is typically used to average over a sequence of tokens. However, the mathematical principle of GRPO is to reduce variance by normalizing advantages within a group.
>
> *   **In Standard GRPO:** The "Group" is a set of $G$ different token sequences generated from one prompt.
> *   **In DRPO:** The "Group" is a set of **$G$ different reward values (actions)** sampled from the policy for one context.
> *   Therefore, DRPO does not operate at a "sequence-level" or "token-level" in the traditional sense; it operates at the **"sample-level"** within the continuous action space of rewards. The GRPO objective is perfectly valid here: we estimate the baseline using the group average of the sampled rewards' meta-values.
>
> We **have revised Appendix B** to explicitly add this discussion.

---

> ### Author Response · Authors · 2025-11-21
> **response to Reviewer jXS5 (2/4)**
>
> ### **Novelty and Distinction from Prior Work**
>
> The reviewer correctly notes that our use of a Normal distribution aligns with prior work [1]. However, we would like to clarify that our core contribution is not the parameterization itself, but the **Reinforcement Learning framework (DRPO)** used to optimize it. We chose the Gaussian distribution as a deliberate design choice to enable this framework, based on three factors:
>
> 1.  **Simplicity & Stability:** The Gaussian is parameterized by only two values ($\mu, \sigma^2$), making the optimization landscape significantly smoother and more efficient than more complex distributions.
> 2.  **Explicit Uncertainty:** The variance $\sigma^2$ provides a direct, interpretable measure of uncertainty. This allows us to analytically derive the **uncertainty-scaled gradients** (Eq. 3 & 4 in the paper), providing a clear theoretical view of how the model learns to suppress updates when uncertain.
> 3.  **Standard Foundation:** It aligns with established distributional methods [1, 2, 3], allowing us to isolate the benefits of our training paradigm.
>
> **The Critical Distinction: RL vs. Supervised Learning**
> The reviewer asks how this improves on existing models. The key distinction lies in **calibration and reliability**. Existing prior distributional models are trained via Supervised Learning (SL). In our early experiments, we observed a critical failure mode in SL which we term **"Variance Collapse."** When trained with standard SL objectives (like Regression or Bradley-Terry), the model’s variance $\sigma$ often shrinks rapidly to near-zero values to minimize training error. This effectively reduces the distributional model to a standard point-estimate regressor, stripping it of the ability to express meaningful uncertainty or perform calibration. We **have added this early trial and discussion to Appendix D**.
>
> **How DRPO Solves This:**
> Our **Contextual Bandit** formulation prevents this collapse through the **Exploration-Exploitation trade-off**, a mechanism unique to RL that is absent in SL:
>
> *   **Exploration (Maintaining Variance):** In DRPO, the reward outputs are actions sampled from the policy $\pi_\theta(r|x,y) \sim \mathcal{N}(\mu, \sigma^2)$. To receive high meta-rewards, the agent *must* explore. If $\sigma$ collapses to zero prematurely, the agent loses its ability to "search" for the optimal reward regions that satisfy the margin condition.
> *   **Exploitation (Principled Calibration):** The model effectively learns to reduce $\sigma$ (exploit) *only* when it has found a region where the preference margin is statistically significant.
>
> This active search results in a model that is not just accurate, but **well-calibrated**, which indicates **knowing when it is likely to be wrong**. This is empirically evidenced by the strong linearity of our calibration curves in Figure 4b, a result that standard SL methods struggle to achieve due to variance collapse.
>
> [1] Lou, Xingzhou, et al. "Uncertainty-aware reward model: Teaching reward models to know what is unknown." arXiv preprint arXiv:2410.00847 (2024).
>
> [2] Yan, Yuzi, et al. "Reward-robust rlhf in llms." arXiv preprint arXiv:2409.15360 (2024).
>
> [3] Sun, Wangtao, et al. "Probabilistic Uncertain Reward Model." arXiv preprint arXiv:2503.22480 (2025).

---

> ### Author Response · Authors · 2025-11-21
> **response to Reviewer jXS5 (3/4)**
>
> ### **Explanation of Hyperparameters (Sensitivity of $K$)**
>
> We thank the reviewer for highlighting the importance of the confidence-controlling hyperparameter $K$.
>
> **The Dual Role of Uncertainty in DRPO:**
> To fully explain the impact of $K$, we must first clarify that our framework incorporates uncertainty in two distinct places:
>
> 1.  **Intrinsic (Gradient Dynamics):** As analyzed in **Section 4.3 (Eq. 3)**, the policy gradient for the mean $\mu_\phi$ is naturally scaled by the inverse variance $1/\sigma_\phi^2$. This acts as an automatic, adaptive learning rate—suppressing updates when the model is uncertain and amplifying them when confident. **This mechanism is intrinsic to the distributional policy gradient and remains active regardless of $K$.**
> 2.  **Explicit (Meta-Reward Signal):** The hyperparameter $K$ controls the **meta-reward signal** (Definition 1). It sets the "dynamic margin" $m_{ij}$ proportional to the standard deviation. This ensures the model is only rewarded for preference differences that are **statistically significant** relative to the current noise level. **Intuition:** A larger $K$ makes the meta-reward sparser and stricter,  where the model is only rewarded if it distinguishes the preferred/dispreferred responses with high statistical significance (relative to its current uncertainty). A smaller $K$ provides denser signals but may reward noise.
>
> **Sensitivity Analysis:**
>
> To address the reviewer's concern regarding the design choice of the margin $m_{ij}$, we conducted an additional ablation study on the confidence hyperparameter $K$. We varied $K \in \{0, 0.1, 0.3, 0.5, 1.0\}$ and evaluated the average performance across five established benchmarks (excluding PPE-based benchmarks due to their long time cost and computational constraints within the rebuttal window).
>
> The results, showed in **Figure 7 (in Appendix D)**:
> 1.  **$K=0$ (Baseline):** **Even with $K=0$, the model performs well due to the *Intrinsic* gradient scaling mentioned above.**  However, without the margin, the meta-reward credits "lucky guesses" where distributions overlap significantly, preventing the model from reaching peak performance.
> 2.  **$K \in [0.1, 0.3]$ (Optimal Region):** Performance peaks at $K=0.1$ and remains robust at $K=0.3$. This range effectively filters out statistical noise without discarding valid signals, striking the ideal balance for learning robust features.
> 3.  **$K > 0.5$ (Over-Regularization):** Performance degrades as $K$ increases further. An overly strict margin creates a **sparse reward signal**—the policy struggles to find samples that satisfy the condition $r^w - r^l > m_{ij}$, slowing down convergence. Furthermore, forcing such a large margin on the training set can lead to overfitting, harming generalization to unseen test data.
>
>
>
>
>
>
>
> ### **Q1: Why the standard normal distribution as a fixed prior?**
>
> We initialize with and regularize towards $\mathcal{N}(0,1)$ because it represents an **unbiased, uninformative prior**. Before training, the model should not assume any specific reward scale or preference direction. This prevents the reward values from drifting to arbitrary scales (e.g., exploding to infinity) during RL training, acting as a necessary anchor for the policy.

---

> ### Author Response · Authors · 2025-11-21
> **response to Reviewer jXS5 (4/4)**
>
> ### **Q2: The Two-Stage Method and Objective Function Details**
>
> **Motivation for the Two-Stage Curriculum (DRPO-Zero vs. DRPO-Refine)**
> The reviewer asks about the justification for the two-stage approach and the difference between Eq. (1) and Eq. (2). We employ a curriculum learning strategy designed to ensure stability and, crucially, **calibration**.
>
> *   **The Failure of Standard SFT Initialization (Variance Collapse):**
>     A common paradigm in RLHF is "SFT first, then RL." In our early experiments, we attempted to initialize our reward model using supervised learning (regression or Bradley-Terry loss) before applying DRPO-Refine. However, we encountered a critical issue: **"Variance Collapse."** Because SL objectives strictly minimize prediction error, the model's variance parameter $\sigma$ rapidly shrank to near-zero values. This created a "dead" model for the subsequent RL stage. With $\sigma \approx 0$, the policy had no exploration capability, effectively freezing the learning process. We **have added this early trial and discussion to Appendix D**.
>
> *   **Why DRPO-Zero (Stage 1):**
>     To avoid this, Stage 1 (DRPO-Zero, Eq. 1) trains via RL from scratch (starting from a neutral $\mathcal{N}(0,1)$ prior). It uses looser constraints (lower $K$, lower KL penalty) to allow the model to explore and learn the "rough shape" of the reward landscape while **maintaining a healthy variance** for exploration.
>
> *   **Why DRPO-Refine (Stage 2):**
>     Once the foundational model is learned, Stage 2 (Eq. 2) refines it. We switch the reference policy $\pi_{ref}$ to the learned Stage 1 model and tighten the constraints (higher $K$, higher $\beta'$). This forces the model to focus on "hard" samples and refine its uncertainty estimates. This curriculum is essential for achieving the high calibration shown in Figure 4, which a single stage struggled to achieve.
>
> **Clipping and Importance Sampling**
> The reviewer asks why clipping and importance sampling (standard in PPO/GRPO) are not emphasized or necessary in our objective. This is due to a fundamental difference in computational cost between **token generation** and **reward generation**:
>
> *   **Standard LLM-RL (PPO/GRPO):** Generating text sequences is computationally expensive. Therefore, standard methods generate samples once (using $\pi_{old}$) and perform multiple gradient updates on that same batch. This causes the policy $\pi$ to drift away from $\pi_{old}$, necessitating Importance Sampling ratios $\rho$ and Clipping to correct for the distribution shift and ensure stability.
> *   **DRPO:** Generating a scalar reward value is computationally trivial. This allows us to sample **strictly on-policy** at every single optimization step.
>     *   Since we resample data from the current policy $\pi_\theta$ for every update, our sampling policy is identical to our optimization policy: $\pi_{old} = \pi_\theta$.
>     *   Consequently, the importance ratio is always $\rho_t(\theta) = \frac{\pi_\theta(a|s)}{\pi_{old}(a|s)} = 1$.
>     *   With $\rho=1$, the clipping term naturally becomes redundant. This simplifies the algorithm to a pure policy gradient update without the need for the complexity of off-policy corrections.
>
> We **have added a clarification in Section 4.2** to highlight this efficiency advantage of stateless reward optimization.
>
>
>
> We hope that our responses and the corresponding manuscript revisions have addressed your concerns. We welcome any further questions and would be happy to provide any additional clarification needed.

---

> ### Author Response · Authors · 2025-11-27
>
> Dear Reviewer jXS5,
>
> We sincerely thank you again for your time and the constructive feedback provided in your initial review.
>
> We wanted to follow up to see if you have had a chance to review our detailed response and the revised manuscript. In our rebuttal, we have specifically addressed your primary concerns regarding:
>
> 1.  **The Contextual Bandit Formulation:** We clarified the distinction between token-level generation (MDP) and our reward-generation approach (Bandit), explaining why the stateless formulation is theoretically sound.
> 2.  **Novelty & Training Dynamics:** We elaborated on why the RL-based optimization prevents the "Variance Collapse" often seen in supervised distributional models, and how the Gaussian parameterization enables uncertainty-aware gradient scaling.
> 3.  **Objective Function Details:** We clarified the use of clipping and importance sampling in the context of on-policy scalar sampling.
>
> We believe these clarifications and the additional experimental details in the revision effectively address the issues of soundness and presentation.
>
> As the discussion period is drawing to a close, we would greatly appreciate it if you could take a moment to re-evaluate our paper based on these updates. We remain fully available to answer any further questions or engage in additional discussion.
>
> Best regards,
>
> Authors of Paper 9353

---

> > ### Comment · Reviewer_jXS5 · 2025-11-27
> >
> > Thanks for the detailed response, and I appreciate the efforts made to improve the manuscript. The DRPO-zero is an interesting design, but  I have a differing perspective regarding a few points.
> >
> > In my opinion, I feel the claim  “Exploration-Exploitation” is somewhat misplaced. This concept is usually associated with online settings, while the work here focuses on an offline setting.  Second, framing DRPO-zero through the lens of a contextual bandit problem creates some ambiguity. As the DRPO-zero is essentially a maximum likelihood estimation with a penalty, discussing it from that perspective might be more intuitive, since the core of the first stage relies on the design of Meta-Reward. The use of the contextual bandit formulation to describe this was a source of confusion for me during the previous review phase. Third, I also find the term 'two-stage RL' to be slightly imprecise, as neither stage fits the strict definition of an MDP.
> >
> > Another question is about the experiments. Due to time constraints, I was unable to review the experimental section in full detail. Could you kindly point me to any results that demonstrate the benefit of the proposed reward model for downstream tasks like RLHF or RLVR?
> >
> > I look forward to the authors' feedback and remain open to raising my initial score based on your clarifications.

---

> ### Author Response · Authors · 2025-11-27
> **Clarifying the RL Formulation and Downstream Benefits**
>
> We sincerely thank the reviewer for their continued engagement and for acknowledging the improvements to the manuscript. We would like to address your remaining conceptual concerns and point you to the specific experimental evidence you requested.
>
> ### **Why DRPO is Reinforcement Learning (Contextual Bandit), not MLE**
>
> The reviewer suggests that our method might be better described as "MLE with a penalty" because the context data is offline. We respectfully argue that the **Contextual Bandit formulation is the mathematically precise definition** of our optimization process, and the "Exploration-Exploitation" trade-off is indeed the core mechanism driving the learning.
>
> **A. Our Policy Models Rewards, Not Text (Bandit vs. MDP)**
>
> The confusion regarding MDPs likely stems from standard text generation.
>
> *   **Standard Text Gen:** The policy generates a sequence of tokens. State transitions exist. This is an MDP.
> *   **DRPO:** Our policy $\pi_\phi$ takes a Context (prompt+response) and outputs a **single Action (a reward score/distribution)**.
>     *   There is **no next token, no sequence**, and **no state transition**. The action is terminal.
>     *   A reinforcement learning problem with a horizon of $T=1$ is, by definition, a **Contextual Bandit**.
>     *   **Framing this as an MDP would be technically incorrect**, while framing it as MLE would ignore the dynamic search process.
>
> **B. "Offline" Contexts vs. "On-Policy" Actions (The Active Agent)**
>
> The reviewer notes that we use "offline data." It is crucial to distinguish between the *Environment* and the *Agent*:
>
> *   **Fixed Environment:** The contexts (input preference text) are indeed drawn from a static dataset. **This is standard practice in almost all RLHF/RLVR methods** (e.g., PPO/GRPO for text generation is trained on fixed prompt sets).
> *   **On-Policy Actions:** Crucially, the reward scores (actions) are **not** retrieved from a dataset. In every training step, our policy $\pi_\phi$ **actively samples** fresh reward values $r \sim \mathcal{N}(\mu, \sigma^2)$. The agent interacts with the meta-reward function dynamically.
> *   **The RL Loop:** The training follows the standard RL interaction loop: **Context $\to$ Action (On-policy Sampled) $\to$ Meta-Reward (Calculated) $\to$ Update (via Policy Gradient)**. This dynamic generation of actions is what defines it as **On-Policy RL**.
>
> **C. Why "Exploration" is Essential (Not MLE)**
>
> If this were simply "MLE with a penalty," the model would regress to a static target. In DRPO, the model must **explore**:
>
> *   The policy must maintain variance to find reward values that satisfy the meta-reward metric.
> *   If the model stops exploring (variance collapse), it fails to find these regions.
> *   The optimization balances **expanding variance** (Exploration) against **contracting variance** (Exploitation). This is the classic **Exploration-Exploitation trade-off**, which is fundamentally absent in MLE.
>
> To address similar questions from Vphz regarding the distinction between Bandits and MDPs in this setting, we have added a detailed discussion in **Appendix B.1: DISCUSSION ON RL FORMULATION**.
>
> ### **Benefits for Downstream Tasks (RLHF/RLVR)**
>
> The reviewer asks a valuable question regarding the direct downstream impact of using DRPO for policy fine-tuning. While we did not perform a full PPO/GRPO training run within the rebuttal window, we respectfully highlight that our evaluation on **RewardBench v2** and **RMB** directly assesses this capability.
>
> Specifically, both RewardBench v2 and RMB explicitly evaluate **Best-of-N (BoN)** performance—the ability of the reward model to correctly identify the highest-quality response among multiple model-generated candidates.
>
> *   **Proxy for Fine-tuning:** This BoN capability is not just a static metric; it is the fundamental operation behind **Rejection Sampling Fine-tuning** and is highly correlated with the signal quality required for **Reinforcement Learning Fine-tuning**. If a reward model can reliably rank a set of $N$ candidates, it provides the necessary gradient signal for policy improvement.
> *   **Empirical Evidence:** As shown in **Table 2**, DRPO demonstrates superior performance on these benchmarks. For instance, on **RewardBench v2**, `DRPO-Qwen3-8B` achieves a score of **70.0**, significantly outperforming the much larger `InternLM2-20B` (56.3).
>
> We are happy to answer any further questions.
>
> Best regards,
> Authors of Paper 9353

---

> > ### Comment · Reviewer_jXS5 · 2025-11-28
> >
> > Thanks for the clarification, which addresses most of my concerns. I will raise my score to 6 to reflect these efforts.

---

> > > ### Author Response · Authors · 2025-11-28
> > >
> > > We sincerely thank you for your engagement throughout the rebuttal process and for your decision to raise the score.
> > >
> > > We are glad that our clarifications regarding the Contextual Bandit formulation and the downstream benefits addressed your concerns. Your constructive feedback has pushed us to significantly strengthen the presentation of the paper. We will ensure these clarifications remain in the revised manuscript to improve clarity for future readers.
> > >
> > > Best regards,
> > >
> > > Authors of Paper 9353

---

### Official Review · Reviewer_3euk · 2025-10-31

**Soundness:** 3
**Presentation:** 2
**Contribution:** 3
**Rating:** 6
**Confidence:** 3

**Summary:**

**The paper *“Learning to Reward: A Contextual Bandit Framework for Distributional Reward Policy Optimization”*** proposes **Distributional Reward Policy Optimization (DRPO)**, a two-stage RL-style training framework for reward models used in large language model (LLM) alignment.

The authors start from the observation that current reward model (RM) work mostly fills three quadrants of a \\( 2 \times 2 \\) taxonomy:

\\[
\\begin{array}{c|cc}
 & \\textbf{Supervised} & \\textbf{Reinforcement Learning (RL)} \\\\ \\hline
\\textbf{Point-Estimate} &
\\text{Existing Work} &
\\text{Existing Work} \\\\[2ex]
\\textbf{Distributional} &
\\text{Existing Work} &
\\boxed{\\textbf{--- Missing Quadrant (DRPO) ---}} \\\\
\\end{array}
\\]

The missing quadrant is the **distributional + RL** setting.

DRPO is designed to fill exactly this gap: it treats the reward model itself as a **stochastic policy** in a **contextual bandit** setting, where the “action” is the reward value (drawn from a learned distribution) assigned to a prompt–response pair, thereby **explicitly quantifying uncertainty** in the reward assignment process.

The core technical piece is an **uncertainty-aware meta-reward** (Definition 1) that compares samples from the preferred and dispreferred reward distributions but only gives credit when the difference exceeds a **variance-dependent margin**. This design prevents reward hacking via variance inflation and makes the gradient **uncertainty-aware**.

DRPO is trained in two curriculum stages:

1. **DRPO-Zero**: initialized from a standard normal prior and regularized by KL divergence to \( \mathcal{N}(0,1) \).
2. **DRPO-Refine**: initialized and KL-anchored to the DRPO-Zero model, with stricter margins and stronger KL regularization.

Experiments on seven RM benchmarks using **LLaMA-3.1-8B**, **Qwen3-8B**, and **Qwen3-4B** backbones show that RL–distributional DRPO is **competitive with or better than** strong supervised (SL) point-estimate and SL-distributional baselines, sometimes matching larger models.

The uncertainty-aware meta-reward successfully avoids the **variance-hacking failure mode** observed in simpler hinge-based meta-rewards. The paper also provides a **gradient-level comparison** to the Bradley–Terry model, showing that DRPO naturally downweights low-confidence cases.

Parts of this review were discussed with a colleague to ensure clarity and accuracy.

**Strengths:**

### Novelty
A major strength of this work is its clear identification of a missing quadrant in the
$2 \times 2$ taxonomy of reward modeling: \emph{distributional} vs. \emph{point-estimate}
and \emph{supervised} vs. \emph{RL}. While prior studies cover the other three quadrants,
the \textbf{distributional + RL} setting remained unexplored. DRPO is the first framework
to fill this gap, enabling explicit \textbf{uncertainty modeling} through its stochastic
policy formulation. The framing is compelling, though the contribution would be more robust
if the paper more clearly articulated \textbf{why reinforcement learning is necessary}
beyond supervised distributional approaches.

### Uncertainty-Aware Meta-Learning Formulation

Another key strength lies in the paper’s proposed \textbf{uncertainty-aware meta-learning}
framework. By comparing sampled rewards from preferred and dispreferred responses and scaling
the update by a variance-dependent margin, the method adapts learning to the model’s own
confidence. This prevents reward hacking through variance inflation and yields more stable
gradients. The formulation effectively couples \textbf{meta-learning and uncertainty modeling},
allowing the reward model to self-calibrate and improve robustness across preference data.

The power of this method is evident in ablations.


Additionally, the paper provides a clear \textbf{theoretical comparison with the Bradley–Terry (BT)}
model, showing that DRPO generalizes BT by incorporating variance as a learned parameter.
This analysis strengthens the conceptual grounding of DRPO, illustrating how it extends classical
reward modeling frameworks to explicitly reason about uncertainty.


### Experiments
DRPO’s empirical results represent a clear strength of the work. Evaluated across seven diverse reward modeling benchmarks, DRPO consistently matches or outperforms strong supervised point-estimate and distributional baselines, rivaling larger models. Notably, it performs well across different model architectures, demonstrating strong \textbf{cross-architecture generalization}. These achievements highlight the \textbf{scalability, generality, and practical effectiveness} of combining distributional reward modeling with reinforcement learning, providing strong empirical validation for the proposed framework.

**Weaknesses:**

### Guassian Assumption

A key limitation of DRPO is its reliance on a **Gaussian assumption** for modeling reward distributions.
While this simplifies training and allows for a closed-form uncertainty term, it limits expressiveness—real preference
data can be asymmetric or multimodal. This assumption may therefore yield miscalibrated uncertainty estimates,
reducing robustness in more complex or non-Gaussian reward settings.

### Explanation of RL

While DRPO effectively motivates the need for **distributional** reward modeling,
it provides less clarity on why **reinforcement learning** is necessary over
supervised alternatives. The paper frames the problem as a contextual bandit and
uses RL to optimize a stochastic reward policy, but it does not convincingly
demonstrate where supervised training fails or how RL uniquely improves performance.
A stronger justification for the RL component would make the framework’s design
more compelling and conceptually grounded.

### Experiments

All experiments are trained on a single dataset.

### Presentation

The paper contained minor grammatical errors and **Table 1** was difficult to read because of the formatting of the citations.

**Questions:**

Why is reinforcement learning strictly necessary for the proposed formulation—could supervised distributional training achieve similar outcomes?

---

> ### Author Response · Authors · 2025-11-21
> **response to Reviewer 3euk (1/3)**
>
> We thank the reviewer for the positive assessment of our work, specifically for recognizing the novelty of filling the missing "Distributional + RL" quadrant and the effectiveness of our uncertainty-aware meta-learning formulation. We appreciate the constructive feedback regarding the Gaussian assumption and the motivation for RL.
>
> Below, we address the specific concerns and the core question raised.
>
> ### **W1: Guassian Assumption.**
>
>  We appreciate the reviewer’s insight regarding the potential asymmetry or multimodality of preference data. While we acknowledge that a Gaussian is a simplification, we emphasize that it represents a significant leap in expressiveness compared to standard scalar (point-estimate) reward models, which are entirely incapable of capturing aleatoric uncertainty.
>
> Our choice was based on three main factors:
>
> 1.  **Simplicity & Stability:** The Gaussian is parameterized by only two values (mean $\mu$ and variance $\sigma^2$). This makes the optimization landscape significantly smoother and more efficient to train.
> 2.  **Explicit Uncertainty:** The variance ($\sigma^2$) provides a direct and interpretable measure of predictive uncertainty. This is the cornerstone of our **Uncertainty-Aware Meta-Reward** (Eq. 1) and our calibration analysis. A Gaussian allows us to analytically derive the **uncertainty-scaled gradients** (Eq. 3 & 4), providing a clear theoretical view of how the model learns.
> 3.  **Strong Foundation:** It is a standard and effective choice in related distributional methods [1, 2, 3], providing a solid baseline to validate our novel RL-based training scheme.
>
> **New Experiment with Categorical Distribution:** To address the concern regarding limited expressiveness, we implemented a variant of DRPO using a **Categorical distribution** (modeling the reward as a probability distribution over discrete value bins). Due to the time constraints of the rebuttal, we performed minimal hyperparameter tuning and evaluated on 5 of the 7 benchmarks (excluding PPE tasks due to their high computational cost).
>
> **The results (detailed in Appendix D.3)** are summarized below:
>
> | Model                  | RewardBench |  RB v2   |   RMB    | RM-Bench | JudgeBench | **Avg**  |
> | :--------------------- | :---------: | :------: | :------: | :------: | :--------: | :------: |
> | **DRPO (Gaussian)**    |  **87.6**   | **68.8** | **64.4** |   69.2   |    63.1    | **70.6** |
> | **DRPO (Categorical)** |    82.6     |   65.3   |   62.3   | **69.9** |  **65.7**  |   69.2   |
>
> **Analysis:**
>
> 1.  **Comparability:** Despite limited tuning, the Categorical variant achieves performance comparable to the Gaussian model (Avg 69.2 vs. 70.6), notably outperforming the Gaussian model on RM-Bench and JudgeBench.
> 2.  **Framework Robustness:** This result confirms that t**he effectiveness of DRPO stems from the Contextual Bandit formulation (the exploration-exploitation dynamics), rather than being dependent on a specific distributional assumption.** The framework is flexible enough to accommodate more expressive distributions when needed.
>
> **We have added these results and a detailed analysis to Appendix D.3.**
>
> [1] Lou, Xingzhou, et al. "Uncertainty-aware reward model: Teaching reward models to know what is unknown." arXiv preprint arXiv:2410.00847 (2024).
>
> [2] Yan, Yuzi, et al. "Reward-robust rlhf in llms." arXiv preprint arXiv:2409.15360 (2024).
>
> [3] Sun, Wangtao, et al. "Probabilistic Uncertain Reward Model." arXiv preprint arXiv:2503.22480 (2025).

---

> ### Author Response · Authors · 2025-11-21
> **response to Reviewer 3euk (2/3)**
>
> ### **W2 and Q1: Why RL.**
>
> The reviewer asks a critical question: *Could supervised distributional training achieve similar outcomes?*
>
> Our answer is: **Yes, for accuracy, but No, for calibration and uncertainty quantification.**
>
> **Accuracy vs. Calibration (The Limitation of SL)**
> As shown in Table 2, we agree that Supervised Learning (SL) distributional methods (like URM) can achieve competitive performance on standard benchmarks. If the goal were solely to maximize reward accuracy on in-distribution data, SL might suffice.
>
> However, the primary motivation for Distributional Reward Modeling is not just accuracy, but **reliability, uncertainty quantification, and calibration**. It is here that SL approaches often fall short. In our early experiments, we attempted to train the Gaussian distribution using a standard Supervised Bradley-Terry (BT) loss as Stage 1. We observed a phenomenon of **"Variance Collapse,"** where the model's variance $\sigma$ rapidly shrank to near-zero values. **This collapse effectively reduced the distributional model to a point-estimate regressor**, stripping it of the ability to express uncertainty or perform the calibration shown in **Figure 4**. We **have added this early trial and discussion to Appendix D**.
>
> **Why RL Prevents Collapse (The Contextual Bandit Formulation)**
> Our **Contextual Bandit** formulation prevents this collapse and enforces calibration through the **Exploration-Exploitation trade-off**:
>
> *   **Exploration (Maintaining Variance):** The policy $\pi_\theta(r|x,y) \sim \mathcal{N}(\mu, \sigma^2)$ must explore. If $\sigma$ collapses to zero too early, the agent stops exploring and fails to find the optimal reward regions that satisfy the meta-reward margin. The RL objective implicitly penalizes premature variance collapse because a deterministic policy cannot "search" for the margin.
> *   **Exploitation (Learning Confidence):** The model exploits by shifting $\mu$ and adjusting $\sigma$ *only* when it finds a region where the preference margin ($r^w > r^l$) is statistically significant.
> *   **Active Search vs. Passive Fitting:** Unlike SL, which passively fits a label (often overfitting to the noise), the RL agent actively searches for a distribution that satisfies the uncertainty-aware meta-reward. This results in a model that is not just accurate, but **well-calibrated**—knowing when it is likely to be wrong (as evidenced by the linear calibration curves in Figure 4b).

---

> ### Author Response · Authors · 2025-11-21
> **response to Reviewer 3euk (3/3)**
>
> ### **W3: Trained on a Single Dataset.**
>
> We apologize if the description of our training data was unclear. While we refer to our training source as the "Skywork-80K" dataset, it is important to clarify that **this is not a single, narrow-domain source.**
>
> **Diversity of Skywork-80K:**
> As detailed in the original Skywork-Reward paper [1], Skywork-80K is a high-quality **composite dataset** constructed by aggregating and filtering diverse preference sources. It explicitly includes data from:
>
> * **HelpSteer2 [2]:** High-quality helpfullness and steerability data.
>
> * **OffsetBias [3]:** Data specifically designed to mitigate length/format bias.
>
> * **WildGuardMix [4]:** Focus on safety and refusal handling.
>
> * **Magpie Ultra / Pro / Air [5]:** Large-scale synthetic instruction tuning data covering reasoning and general chat.
>
>   Therefore, our model was effectively trained on a multi-source mixture covering reasoning, safety, and general conversation, rather than a single narrow domain.
>
> **Out-of-Distribution (OOD) Generalization:**
> This diversity in training data, combined with our DRPO method, explains why our model **generalizes so well to the seven distinct benchmarks** shown in Table 2. These benchmarks contain prompt distributions significantly different from any single subset of Skywork-80K, yet DRPO achieves state-of-the-art or competitive performance. This confirms that the model has learned robust, generalized preference representations rather than overfitting to a specific dataset artifact.
>
> We  **have updated Appendix C** to explicitly list the constituent datasets of Skywork-80K to highlight the diversity of our training data.
>
> [1] Liu, Chris Yuhao, et al. "Skywork-reward: Bag of tricks for reward modeling in llms." arXiv preprint arXiv:2410.18451 (2024).
>
> [2] Wang, Zhilin, et al. "Helpsteer 2: Open-source dataset for training top-performing reward models." Advances in Neural Information Processing Systems 37 (2024): 1474-1501.
>
> [3] Park, Junsoo, et al. "Offsetbias: Leveraging debiased data for tuning evaluators." Findings of the Association for Computational Linguistics: EMNLP 2024. 2024.
>
> [4] Han, Seungju, et al. "Wildguard: Open one-stop moderation tools for safety risks, jailbreaks, and refusals of llms." Advances in Neural Information Processing Systems 37 (2024): 8093-8131.
>
> [5] Xu, Zhangchen, et al. "Magpie: Alignment data synthesis from scratch by prompting aligned llms with nothing." arXiv preprint arXiv:2406.08464 (2024).
>
> We hope that our responses and the corresponding manuscript revisions have addressed your concerns. We welcome any further questions and would be happy to provide any additional clarification needed.

---

> ### Author Response · Authors · 2025-11-27
>
> Dear Reviewer 3euk,
>
> We sincerely thank you again for your positive assessment and your insightful questions regarding the necessity of the RL formulation and the Gaussian assumption.
>
> We have posted a detailed response and updated our manuscript to address your concerns. With the discussion period concluding in a few days, we would be very grateful if you could check whether these new results and clarifications address your concerns. We are more than happy to provide further details if any questions remain.
>
> Best regards,
>
> Paper 9353 Authors

---

### Official Review · Reviewer_Vphz · 2025-11-03

**Soundness:** 2
**Presentation:** 3
**Contribution:** 2
**Rating:** 4
**Confidence:** 3

**Summary:**

This paper introduces Distributional Reward Policy Optimization (DRPO), a framework that formulates the training of uncertainty-aware, distributional reward models as a contextual bandit problem. The approach uses reinforcement learning, treating the reward model as a stochastic policy and optimizing it with an uncertainty-aware meta-reward signal. A two-stage curriculum is employed: DRPO-Zero (prior-guided pre-training) and DRPO-Refine (self-refinement), with extensive empirical evaluation across multiple benchmarks for language model alignment and comprehensive algorithmic analysis.

**Strengths:**

1. The authors provide a thorough mathematical analysis of the proposed framework, comparing DRPO's gradient dynamics to standard point-estimate approaches and offering explicit derivations (see Section 4.3 and equations for the gradient derivations).

2. The uncertainty-aware meta-reward design is justified both intuitively and mathematically. Experiments and ablations in Figures 3a-3d  illustrate that DRPO's method avoids pathological increases in variance and reward hacking seen in naive hinge-based approaches-this is particularly compelling as it anchors the value of the proposed uncertainty calibration.

**Weaknesses:**

1. **Empirical Performance Margins Are Modest.** While Table 2 shows that DRPO is highly competitive, performance gains over the best prior methods are relatively incremental or size-dependent. For many tasks, DRPO is indistinguishable from, or slightly behind, strong SL distributional baselines like URM-Skywork-8B or Skywork-Llama3.1-8B. In the 4B configuration, DRPO does not consistently outperform the best prior models. The claimed advantage would be more persuasive with statistical significance tests, effect size reporting, or robust anomaly analysis.
2. **The claim of filling the (RL, Distributional) quadrant is weak.** The model is trained entirely offline on a fixed dataset, eliminating the fundamental exploration-exploitation trade-off required in true contextual bandit problems. The "policy gradient" is merely a term for a complex, sampling-based loss calculation, not a true interactive learning process. The framework is more accurately described as an advanced (SL + Distributional) method, making the "RL" claim tenuous
3. **The paper adopts a Gaussian distribution to model human preferences without adequately justifying this choice.** A Gaussian is unimodal and cannot capture the complex, often multi-modal (e.g., polarized) nature of human preferences. The paper fails to discuss why this restrictive model is superior to more flexible alternatives (like Categorical distributions) mentioned in the (SL, Distributional) context.

**Questions:**

1. Could the authors elaborate on why a Gaussian was chosen over more flexible alternatives, such as a Categorical distribution (which is noted in the SL quadrant).

2. A more direct and insightful ablation would be to test the framework using the exact formulation from Definition 1, but simply setting $K=0$. This would create a sampling-based baseline that is "uncertainty-agnostic." Can the authors provide experimental results for this $K=0$ baseline? This would more directly demonstrate the importance of the $\sigma_{diff}$ term in stabilizing training and preventing the "variance inflation" seen in the hinge-based model.

---

> ### Author Response · Authors · 2025-11-21
> **response to Reviewer Vphz (1/3)**
>
> We sincerely thank Reviewer Vphz for the detailed and insightful feedback. We are encouraged that the reviewer found our mathematical analysis thorough and our uncertainty-aware meta-reward design compelling. We address the specific weaknesses and questions below:
>
> ### **W1: Empirical Performance Margins are Modest.**
>
> We agree that while DRPO is highly competitive, it does not uniformly dominate all baselines across all seven benchmarks. Our primary contribution is not simply achieving a new state-of-the-art, but introducing **a robust and well-calibrated RL-based paradigm for distributional reward modeling**. We wish to highlight three key aspects of our performance: 1) **Efficiency vs. Scale:** While raw gains on some benchmarks are modest, we highlight the parameter efficiency of DRPO. As shown in Table 2, DRPO-Qwen3-4B model competes with and often outperforms significantly larger baselines, such as InternLM2-20B and ArmoRM-Llama3-8B. 2) **Superior Calibration:** A key advantage of our distributional approach, not captured by accuracy scores alone, is superior model calibration. As shown in Figure 4b, the DRPO-Refine model's predictive confidence closely tracks its empirical accuracy. This is a critical feature for reliability and for downstream applications like active learning or out-of-distribution detection. 3) **Contribution Beyond Raw Scores:** We believe the main benefit of DRPO lies in its training dynamics and the quality of the learned model, not just its final accuracy. As the reviewer noted, our uncertainty-aware meta-reward prevents the reward-hacking failure mode seen in naive baselines (Figures 3a-d).
>
> ### **W2: The claim of filling the (RL, Distributional) quadrant is weak.**
>
>  We respectfully disagree with the assessment that our framework is "merely an advanced SL method" or that the claim of filling the RL-Distributional quadrant is tenuous. We believe this stems from a misunderstanding of our problem formulation regarding the state and action spaces. We clarify our position through the following three points:
>
> 1. **Contextual Bandits are a Class of Reinforcement Learning**
>
> Our framework mathematically formulates reward modeling as a Contextual Bandit problem. By definition, the contextual bandit is a fundamental reinforcement learning setting where an agent observes a state (context), selects an action, and receives a reward. In DRPO, the "Context" is the prompt-response pair $(x,y)$, and the "Action" is the reward value $r$ generated by the model. Since our method optimizes a policy to maximize a non-differentiable return signal (the meta-reward) rather than minimizing a supervised loss against a ground-truth label, it strictly falls under the definition of Reinforcement Learning.
>
> 2. **Offline Contexts vs. On-Policy and On-line Actions**
>
> The review suggests that using a fixed dataset eliminates the RL nature of the problem. We argue that this conflates the environment setup with the learning paradigm:
>
> - **Standard Practice:** Almost all RLHF methods (e.g., PPO) are trained on fixed datasets of prompts. **The "Offline" nature of the prompts does not make the algorithm Supervised Learning.**
> - **On-Policy and on-line Generation:** In our framework, while the input contexts $(x, y)$ are drawn from a fixed dataset, the **actions (reward scores)** are sampled **on-policy and on-line** from the current distribution $\pi_{\phi}$ in during training.
> - **Analogy:** In standard LLM-RL (e.g., PPO for generation), the input is a query, and the output is a token sequence (response). In DRPO, the input is the pair $(x,y)$, and the output is the score $r$ . In both cases, the output is an action generated by a stochastic policy that must be optimized via policy gradients, not by regression to a target.
>
> 3. **The Exploration-Exploitation Trade-off**
>
> Contrary to the claim that this trade-off is absent, it is central to our meta-reward maximization mechanism:
>
> - **Exploration:** Our policy is modeled as a Gaussian distribution $\mathcal{N}(\mu, \sigma^2)$. During training, the model explores the action space by sampling diverse reward values $r_i$ from this distribution.
> - **Exploitation:** The model exploits by shifting its mean $\mu$ and adjusting its variance $\sigma$ toward regions that yield high **Meta-Rewards** (i.e., regions where the preference margin $r^w > r^l$ is statistically significant).
> - **The Trade-off:** The model must balance increasing variance (exploration) to find samples that might satisfy the margin condition against decreasing variance (exploitation) to ensure consistent high rewards and minimize the KL penalty. Unlike Supervised Learning, which fits a label, our agent actively searches for a distribution of scores that satisfies the uncertainty-aware meta-reward criteria.

---

> ### Author Response · Authors · 2025-11-21
> **response to Reviewer Vphz (2/3)**
>
> ### **W3 & Q1: Justification for the Gaussian Distribution**.
>
> We selected the Gaussian for its practicality and its direct relevance to our core thesis on uncertainty.
>
> *   Our choice was based on three main factors:
>     1.  **Simplicity & Stability:** The Gaussian is parameterized by only two values (mean and variance), making it efficient and stable to train.
>     2.  **Explicit Uncertainty:** The variance (σ²) provides a direct and interpretable measure of predictive uncertainty, which is the cornerstone of our uncertainty-aware meta-reward and our calibration analysis.
>     3.  **Strong Baseline:** It is a common and effective choice in related distributional methods [1,2,3], providing a solid foundation for our RL-based training scheme.
>
> **New Experiment with Categorical Distribution:** To address the concern regarding limited expressiveness, we implemented a variant of DRPO using a **Categorical distribution** (modeling the reward as a probability distribution over discrete value bins). Due to the time constraints of the rebuttal, we performed minimal hyperparameter tuning and evaluated on 5 of the 7 benchmarks (excluding PPE tasks due to their high computational cost).
>
> **The results (detailed in Appendix D.3)** are summarized below:
>
> | Model                  | RewardBench |  RB v2   |   RMB    | RM-Bench | JudgeBench | **Avg**  |
> | :--------------------- | :---------: | :------: | :------: | :------: | :--------: | :------: |
> | **DRPO (Gaussian)**    |  **87.6**   | **68.8** | **64.4** |   69.2   |    63.1    | **70.6** |
> | **DRPO (Categorical)** |    82.6     |   65.3   |   62.3   | **69.9** |  **65.7**  |   69.2   |
>
> **Analysis:**
>
> 1.  **Comparability:** Despite limited tuning, the Categorical variant achieves performance comparable to the Gaussian model (Avg 69.2 vs. 70.6), notably outperforming the Gaussian model on RM-Bench and JudgeBench.
> 2.  **Framework Robustness:** This result confirms that t**he effectiveness of DRPO stems from the Contextual Bandit formulation (the exploration-exploitation dynamics), rather than being dependent on a specific distributional assumption.** The framework is flexible enough to accommodate more expressive distributions when needed.
>
> **We have added these results and a detailed analysis to Appendix D.3.**
>
> [1] Lou, Xingzhou, et al. "Uncertainty-aware reward model: Teaching reward models to know what is unknown." arXiv preprint arXiv:2410.00847 (2024).
>
> [2] Yan, Yuzi, et al. "Reward-robust rlhf in llms." arXiv preprint arXiv:2409.15360 (2024).
>
> [3] Sun, Wangtao, et al. "Probabilistic Uncertain Reward Model." arXiv preprint arXiv:2503.22480 (2025).

---

> ### Author Response · Authors · 2025-11-21
> **response to Reviewer Vphz (3/3)**
>
> ### **Q2:  Ablation with K=0.**
>
> This is an excellent suggestion for a more direct ablation study, and we thank the reviewer for proposing it. The reviewer's framing of this baseline as "uncertainty-agnostic" is insightful, as it highlights a critical distinction in our framework's design. We would like to clarify that **even with K=0, our method is not entirely "uncertainty-agnostic"** due to the nature of our policy gradient. This nuance, in fact, makes the results of this ablation even more compelling.
>
> Our full DRPO framework **incorporates uncertainty in two key places**:
>
> 1.  **In the Learning Update (Intrinsic):** As detailed in our Algorithm Analysis (Section 4.3), the policy gradient for the mean (μ\_φ) is intrinsically scaled by the inverse variance (1/σ\_φ²). This acts as an adaptive learning rate, suppressing updates when the model is uncertain (high σ\_φ) and amplifying them when it is confident (low σ\_φ). This mechanism is always active, regardless of K.
>
> 2.  **In the Learning Signal (Explicit):** The hyperparameter K controls the uncertainty-awareness of the meta-reward, and thus the advantage Â. By setting the dynamic margin m\_ij to be proportional to the standard deviation of the reward difference (K > 0), we ensure the model is rewarded for preferences that are statistically significant.
>
> Setting K=0 disables the second mechanism, isolating the effect of the first. It creates a baseline where the learning *update* is uncertainty-aware, but the learning *signal* (the advantage) is not.
>
> To address the reviewer's concern regarding the design choice of the margin $m_{ij}$, we conducted an additional ablation study on the confidence hyperparameter $K$. We varied $K \in \{0, 0.1, 0.3, 0.5, 1.0\}$ and evaluated the average performance across five established benchmarks (excluding PPE-based benchmarks due to their long time cost and computational constraints within the rebuttal window).
>
> The results, showed in **Figure 7 (in Appendix D)**, reveal a clear performance trajectory that validates our "Signal-to-Noise" interpretation:
>
> 1.  **$K=0$ (Baseline):** **Even with $K=0$, the model performs well due to the *Intrinsic* gradient scaling mentioned above.**  However, without the margin, the meta-reward credits "lucky guesses" where distributions overlap significantly, preventing the model from reaching peak performance.
> 2.  **$K \in [0.1, 0.3]$ (Optimal Region):** Performance peaks at $K=0.1$ and remains robust at $K=0.3$. This range effectively filters out statistical noise without discarding valid signals, striking the ideal balance for learning robust features.
> 3.  **$K > 0.5$ (Over-Regularization):** Performance degrades as $K$ increases further. An overly strict margin creates a **sparse reward signal**—the policy struggles to find samples that satisfy the condition $r^w - r^l > m_{ij}$, slowing down convergence. Furthermore, forcing such a large margin on the training set can lead to overfitting, harming generalization to unseen test data.
>
>
>
> We hope that our responses and the corresponding manuscript revisions have addressed your concerns. We welcome any further questions and would be happy to provide any additional clarification needed.

---

> > ### Comment · Reviewer_Vphz · 2025-11-23
> > **Key Issues Remain Unresolved**
> >
> > Thank you for the authors’ careful and detailed response. However, several key issues remain unresolved.
> >
> > 1. A contextual bandit is strictly a *simplified MDP*, whereas “Reasoning RM” corresponds to a much more general MDP definition.
> >
> > 2. Contextual bandits are inherently an *online decision-making* setting. The fundamental objective is to minimize *online regret* (or equivalently maximize cumulative online reward), and the explore–exploit tradeoff must be grounded in an *online interaction* scenario. I do not see where the online aspect appears in your current setup, since all training is performed entirely on offline data.
> >
> > If you interpret \(R\) as cumulative reward, then a typical *online* contextual bandit loop looks like the following:
> >
> > - The algorithm is given two sets of \(G\) reward samples.
> > - Receives a single contextual input.
> > - Chooses an action using some exploration strategy.
> > - Observes a reward \(r\).
> > - Updates its policy based on the observed reward.
> > - Proceeds to the next contextual instance.
> >
> > If one plots the reward-versus-iteration curve during training, the contextual bandit aims to maximize **the area under the curve**, rather than merely the final converged value. This distinction is essential when evaluating algorithms designed for online settings.
> >
> > I hope these comments help clarify the remaining issues, and I appreciate the authors’ efforts in improving the paper.

---

> > > ### Author Response · Authors · 2025-11-23
> > > **response to "Key Issues Remain Unresolved"**
> > >
> > > We thank the reviewer for the continued engagement. The core of the disagreement may stems from a fundamental mismatch between the reviewer's assumed MDP framework for text generation and our explicit **Contextual Bandit** formulation for **reward modeling**. We will now definitively clarify why our approach is not only valid but is the correct and necessary RL paradigm for this specific task.
> > >
> > > **Our Policy Models Rewards, Not Text. This Mandates a Contextual Bandit Formulation.**
> > >
> > > The central point of confusion appears to be the nature of our policy. Our policy, `π_φ`, is a **reward model**. Its purpose is to take a complete text (`prompt`, `response`) as input and output a **reward distribution**, which is a judgment of that text's quality. It does **not** generate a text sequence.
> > >
> > > *   **No Autoregressive Generation:** Consequently, there is no sequence of actions, no concept of a "next token," and critically, **no state transition**. The action (producing a score) is a terminal, one-step event.
> > > *   **A One-Step RL Problem:** This, by definition, is a one-step reinforcement learning problem. The **Contextual Bandit** framework is the precise mathematical model for such one-step decision-making. Attempting to frame this as an MDP would be conceptually incorrect, as the foundational element of an MDP, i.e. the state transition function, does not exist in our problem setting. The meta-reward is the signal that evaluates the quality of this single scoring action.
> > >
> > > **Our Action Sampling is On-Policy, Even with Offline Contexts.**
> > >
> > > The reviewer's concern about "offline data" seems conflates the input context with the policy's action.
> > >
> > > *   The **contexts** (the `(prompt, response)` pairs to be scored) are indeed from a fixed, offline dataset.
> > > *   However, the **action** (the reward score `r` sampled from our policy `π_φ`) is generated **on-policy** in every single training step.
> > >
> > > The definitive proof of this lies within our own objective function (Eq. 1). Our formulation **does not use importance sampling ratio or the clipping mechanism** common in LLM RL practice like PPO/GRPO for sequence generation. This is because the policy sampling the action is identical to the policy being optimized. The importance sampling ratio is therefore always 1, a defining characteristic of an on-policy algorithm.
> > >
> > > **The "Pure Online" Setting is Infeasible for LLM Alignment.**
> > >
> > > Finally, we must address the reviewer's idealized "online interaction scenario." A fully online setting, where preference data is generated by human labelers in real-time to guide the training loop, is **computationally and logistically infeasible** for training large-scale language models. This is precisely why the entire subfield of LLM alignment, including foundational methods like RLHF and DPO, operates on pre-collected, static preference datasets. Our offline contextual bandit formulation is not a flawed approximation of an online ideal; it is a principled and practical adaptation of RL to the concrete realities of the domain.
> > >
> > > In summary, our framework is a correct, on-policy application of the Contextual Bandit paradigm, which is the appropriate model for the task of learning a reward function. We hope this definitive clarification resolves this issue.  Recognizing that Reviewer jXS5 had a similar concern, we have added **Appendix B.1: DISCUSSION ON RL FORMULATION: CONTEXTUAL BANDITS VS. MDPS** to provide a comprehensive clarification.  Should our explanation remain unclear, or if you have any other concerns, please do not hesitate to let us know. We welcome the opportunity for further discussion.

---

> > > ### Author Response · Authors · 2025-11-24
> > > **additional response to "Key Issues Remain Unresolved"**
> > >
> > > We thank Reviewer Vphz for the thoughtful comments again. We understand the concern regarding the definition of Contextual Bandits (CB) and the distinction between **"online regret minimization" settings** and our **optimization framework**. However, we believe there is a misunderstanding regarding how CB is applied in the context of RLHF versus classical application such as recommendation systems. We clarify these points below:
> > >
> > > **DRPO follows a standard RL interaction loop (On-Policy Training).**
> > >
> > > The reviewer states that our setup lacks the "online aspect" because "all training is performed entirely on offline data." We respectfully clarify that while the inputs (Contexts, prompt-response pairs) come from a fixed dataset, **the agent-environment interaction is fully dynamic and on-policy.**
> > >
> > > As described in Section 4.2 and Section 4.3 of our paper:
> > >
> > > - **Receive Context:** The model receives a  inputs (Contexts, prompt-response pairs) .
> > > - **Choose Action:** The policy $\pi_\phi$ dynamically samples fresh reward values (actions) $r \sim \mathcal{N}(\mu, \sigma^2)$ during training. This is not retrieving offline labels; it is active exploration.
> > > - **Observe Reward:** The environment computes a "Meta-Reward".
> > > - **Update:** The policy is updated via Policy Gradient.
> > >
> > > This process strictly adheres to the **"Context -> Action -> Reward -> Update" loop** cited by the reviewer. The "Online" nature refers to the **On-Policy** generation of actions (rewards) and the immediate feedback loop, which is distinct from "Offline RL" where the agent learns solely from pre-recorded trajectories without exploration.
> > >
> > > **Optimization Objective: Best Policy vs. Cumulative Regret.**
> > >
> > > The reviewer correctly notes that in classical application, such as online recommendation systems, the goal is often to minimize cumulative regret. However, within the RLHF community, the Contextual Bandit formulation is frequently used as an optimization paradigm to maximize the final policy performance, rather than minimizing the cost of exploration during training.
> > >
> > > Our goal, like standard PPO/GRPO in LLMs, is to converge to the optimal reward distribution. The "Contextual Bandit" terminology correctly describes the mathematical structure of the problem (Horizon $T=1$, stateless), as detailed in our Appendix B.1.
> > > We are solving a stateless policy optimization problem. The "Contextual Bandit" framework provides the exact gradient dynamics needed for this (Eq. 3 \& 4). We hope this clarifies that our usage of CB is consistent with Policy Optimization methods, even if the "regret" metric differs from classical RL application.

---

> ### Author Response · Authors · 2025-11-27
>
> Dear Reviewer Vphz,
>
> We would like to sincerely thank you for your continued and detailed engagement with our work. Your thoughtful questions have been invaluable.
>
> We have posted a response to your most recent comments and wanted to ensure you had a chance to see it, as the discussion period is ending in a few days. In our latest response, we aimed to definitively clarify the contextual bandit formulation of our work. Recognizing the importance of this point (which was also raised by Reviewer jXS5), we have consolidated a comprehensive discussion in a new section, Appendix B.1: DISCUSSION ON RL FORMULATION: CONTEXTUAL BANDITS VS. MDPS. We welcome any further discussion.
>
> Thank you again for your time and consideration.
>
> Sincerely,
>
> The Authors of Paper 9353

---

### Author Response · Authors · 2025-11-21

We thank all reviewers for their time, constructive feedback, and insightful comments. We are encouraged that the reviewers appreciated the **novelty of DRPO** in filling the "Distributional + RL" quadrant of reward modeling (Reviewer 3euk), the **theoretical analysis** of the gradient dynamics (Reviewer Vphz), and the **principled uncertainty-aware meta-reward design** that **avoids reward hacking** (Reviewer Vphz, 3euk). We are also glad that the comprehensive experiments demonstrating **cross-architecture generalization** were recognized (Reviewer 3euk, bz4m).

We have carefully considered all suggestions and updated our manuscript accordingly. **Changes in the revised paper are marked in red.** Below, we summarize the major updates and additional experiments conducted during the rebuttal phase:

**1. Clarification on RL Formulation (Appendix B.1 & Section 4.2)**
In response to questions regarding the nature of our RL setup (Reviewer Vphz, jXS5, bz4m), we have added a detailed discussion distinguishing our **Contextual Bandit** formulation from standard MDP-based RLHF. We clarify that:

*   Our problem is **stateless** with a horizon of $T=1$.
*   We utilize **on-policy sampling** at every step, which simplifies the optimization by **removing the need for importance sampling and clipping** often found in RL for language generation.

**2. Justification for Two-Stage RL (Appendix D.1)**
To address why we employ a two-stage RL curriculum instead of a standard SFT+RL approach (Reviewer Vphz, 3euk), we performed an early-trial experiment treating the first stage as Supervised Learning.

*   **Finding:** We observed a phenomenon of **"Variance Collapse,"** where the model's variance $\sigma$ rapidly shrank to near-zero values within the first few steps.
*   **Conclusion:** This collapse effectively reduced the distributional model to a point-estimate regressor, stripping it of the ability to capture uncertainty. This validates that the exploration-exploitation trade-off provided by our bandit formulation is essential even in the initial stage.

**3. Sensitivity Analysis of Confidence Margin $K$ (Appendix D.2)**
We conducted an ablation study on the hyperparameter $K$ (Reviewer Vphz, jXS5). We clarify that **even with $K=0$, our method is not "uncertainty-agnostic."** DRPO incorporates uncertainty in two distinct ways:

*   **Intrinsic (Gradient Scaling):** As shown in our Algorithm Analysis (Eq. 3), the update for the mean $\mu_\phi$ is always scaled by the inverse variance $1/\sigma_\phi^2$. This acts as an adaptive learning rate regardless of $K$.
*   **Explicit (Meta-Reward):** The parameter $K$ controls the strictness of the reward signal. Our results show that while $K=0$ is robust due to intrinsic scaling, a non-zero margin ($K \in [0.1, 0.3]$) further improves performance by explicitly filtering out ambiguous preference pairs.

**4. Generalization to Categorical Distribution (Appendix D.3)**
To demonstrate that DRPO is not limited to Gaussian assumptions (Reviewer Vphz, bz4m), we implemented and evaluated a variant of DRPO using a **Categorical distribution**.

*   **Result:** Despite minimal tuning, the Categorical variant achieved performance comparable to, and on some benchmarks (RM-Bench, JudgeBench) better than, the Gaussian baseline.
*   **Conclusion:** This confirms that the Contextual Bandit formulation of DRPO is robust and effective regardless of the specific parameterization (continuous vs. discrete) of the reward distribution.

We hope these revisions and additional analyses address the reviewers' concerns. We are happy to engage in further discussion.

---

### Meta-Review · Area_Chair_345h · 2026-01-07

**Summary:**

Reviewers acknowledged the novelty of filling the distributional+RL gap but questioned the practical advantage over supervised methods, the RL formulation's rigor, and the Gaussian limitation. The authors responded with additional experiments (categorical distribution, sensitivity analysis, variance collapse study) and clarifications on the bandit formulation.

**Reviewer Concerns:**

Addressed: Gaussian expressiveness (via categorical experiments), meta-reward heuristic (signal-to-noise justification), performance rationale (parameter efficiency, calibration).
Partially addressed: downstream impact (linked to Best-of-N benchmarks).
Outstanding: The debate on whether the offline contextual bandit truly constitutes “RL” per classical online definitions remains unresolved, though authors provided a coherent defense.

**Reviewer Scores:**

Vphz: Initial 4. Unconvinced on RL formulation; score likely unchanged (4).

3euk: Initial 6. Concerns on RL necessity addressed; score may remain unchanged.

jXS5: Initial 4, may be raised to 5 after rebuttal.

bz4m: Initial 4. Additional explanations may raise score to 5.

---

### Decision · Program_Chairs · 2026-01-26

Reject